# Fake drugs: Using Baseline Spectral Fingerprinting and a sorting algorithm to infer quality of medications

Christian Salmon[1], Margaret Salmon[2]*, Marcus Paoletti[2‡], Elaine Xu[2‡], Ronny Priefer[3], Michael Rust[4], Aliea Afnan[1]

1 Center for Global Health Engineering, Western New England University, Springfield, Massachusetts, United States of America, 2 InnovationsCZ, San Francisco, CA United States of America, 3 Massachusetts College of Pharmacy and Health Sciences, Boston, Massachusetts, United States of America, 4 Biomedical Engineering Western New England University, Springfield, Massachusetts, United States of America

☯ These authors contributed equally to this work.
‡ MP and EX also contributed equally to this work.
* margaret@innovationscz.org

**Data Availability Statement:** All relevant data are within the manuscript and its Supporting Information file.

## Abstract

An estimated 30–70% of available medications in low-income countries and conflict states are of low quality or counterfeit. Reasons for this vary but most are rooted in regulatory agencies being poorly equipped to oversee quality of pharmaceutical stocks. This paper presents the development and validation of a method for point-of-care drug stock quality testing in these environs. The method is termed Baseline Spectral Fingerprinting and Sorting (BSF-S). BSF-S leverages the phenomena that all compounds in solution have nearly unique spectral profiles in the UV spectrum. Further, BSF-S recognizes that variations in sample concentrations are introduced when preparing samples in the field. BSF-S compensates for this variability by incorporating the ELECTRE-TRI-B sorting algorithm, which contains parameters that are trained in the laboratory using authentic, proxy low quality and counterfeit samples. The method was validated in a case study using fifty samples that include factually authentic Praziquantel and inauthentic samples prepared in solution by an independent pharmacist. Study researchers were blinded to which solution contained the authentic samples. Each sample was tested by the BSF-S method described in this paper and sorted to authentic or low quality/counterfeit categories with high levels of specificity and sensitivity. In combination with a companion device under development using ultraviolet light emitting diodes, the BSF-S method is intended to be a portable and low-cost method for testing medications for authenticity at or near the point-of-care in low income countries and conflict states.

## Introduction

### Background

It is estimated that 30–70% of available medications in low-income countries and conflict States (LIC/CS) are of low quality or counterfeit (LQ/C) [1–4]. LQ/C medications are a serious

**Funding:** the authors received no specific funding for this work

**Competing interests:** AA is affiliated with Daymark (https://daymarkea.com/). There are no patents, products in development or marketed products associated with this research to declare. This does not alter our adherence to PLOS ONE policies on sharing data and materials.

public health problem that warrants attention given their proliferation and consumption cause failures such as antibiotic resistance, increased morbidity and mortality, and loss of confidence in healthcare systems [5]. The economic and human costs of LQ/C medications are enormous. Annual costs of LQ/C antimalarial drugs from Nigeria alone are an estimated 892$ million with 12,300 excess human deaths [6]. Reasons for the high level of LQ/C vary by country, but almost all are rooted in the fact that manufactured medications are not regulated to the same standard as in resourced economies, resulting in agencies in some countries poorly equipped to oversee the quality of pharmaceutical stocks [3, 7].

For the purposes of this paper, LQ/C medications are defined as either falsified or substandard medications with low concentration (or zero) active pharmaceutical ingredient(s), or medications that contain significant contaminants [8]. Low concentrations can be due to inferior manufacturing or poor point-of-sale storage conditions (exposure to moisture, sunlight, heat, or beyond expiration date). Zero concentration is assumed to be the result of intentional falsification of the product [8, 9]. Contaminants can be a function of poor manufacturing quality control and/or intentional addition of impurities.

Several methods for testing quality of medications already exist with high sensitivity and specificity; however, these generally require laboratory settings that consume large capital/operating costs and technical equipment. These include advanced analytical techniques such as mass spectrometry, high performance liquid chromatography, gas chromatography and nuclear magnetic resonance, none of which are commonly accessible in LIC/CS [10–12].

Lower cost alternatives such as colorimetry and paper chromatography do exist, however, these products have limited specificity and sensitivity. For example, Paper Analytic Devices (PADs) detect the presence of an active pharmaceutical compound, but are not able to determine the concentration of this compound, nor the presence of impurities [13]. The no-cost WHO visual checklist is neither sensitive nor specific [14]. Supply-chain monitoring/security-oriented methods, such as watermarking sealed packages of known inspected drugs are also being developed. However, these necessitate a functional government to provide enforcement oversight and sustained public outreach to educate local healthcare providers and/or consumers as to how a 'quality' package is supposed to appear [11, 12].

## A novel point of care method for testing

This paper presents a novel method for inferring quality of medications, one that is specifically designed for use in LIC/CS at the pharmacy and healthcare provider level (point-of-care). The proof of concept of this method is presented here separately from a companion portable analytic device which is now under development and prototype. Our objective is to design an economically sustainable, highly sensitive and specific point-of-care method that enables authentication of a medication using locally sourced supplies and human technical resources.

The method described below has two companion elements that work in unison towards identifying LQ/C medications. The first is a pharmaceutical compound's unique UV spectrum profile, termed here as Baseline Spectral Fingerprint (BSF). The BSF is used to convert a sample to a numeric equivalent, which is then inputted to the second element: a sorting algorithm (S). The algorithm is used to sort samples being tested as either authentic (meaning presumed high quality) or LQ/C.

To clarify, most compounds such as antimicrobials or other agents have an absorbance profile along the Ultraviolet (UV) light spectrum. This enables UV spectrophotometers to be used to compare the absorbance profile of an unknown sample against a baseline profile for a known authentic high-quality sample. If the unknown sample profile differs from its baseline, this suggests the sample is LQ/C. Conceptually, and the basis of the research presented here,

the entire UV absorbance profile is not necessary for detecting LQ/C in this manner. Instead, a "fingerprint" (the BSF) of the full spectral profile collected at discrete wavelengths sampled from across the profile should suffice to represent an authentic sample as a set of numeric values with the intent to use these values to authentic samples of uncertain provenance collected in the field [15, 16].

Logically, if samples could be perfectly prepared, and absorbance data perfectly captured, then absorbance data from any unknown sample would suffice to compare to a BSF to determine if the sample were authentic or LQ/C. However, it should be expected that samples prepared in the field in LIC/CS using less than ideal equipment will exhibit variability in preparation and concentration of the pharmaceutical compound. This raises the question of just what level of discrepancy relative to a BSF can any random, yet factually authentic, field sample hold while still being correctly defined as authentic or incorrectly as LQ/C.

To mitigate this challenge, the ELECTRE TRI-B sorting algorithm was selected from a class of multicriteria decision analysis tools that are often used to sort or cluster items into user or autonomously defined categories using criteria that are selected to differentiate the items to be sorted or clustered. For this, ELECTRE TRI-B has user defined boundaries between categories, and uncertainty parameters around these boundaries, that enable the system to accommodate some level of uncertainty in the criteria values of the items being sorted. The obvious inference here is that the ELECTRE TRI-B algorithm accommodates for some variability in sample preparation in less than ideal field-testing circumstances.

In the specific instance of this paper, the items to be sorted are the samples of medications being prepared for testing. The criteria used to sort these items are the measures of absorbance captured at wavelengths in the ultraviolet (UV) light spectrum, and the sorting system as defined by the BSF-S for a specific medication of interests. This BSF-S method is described and demonstrated below first as a conceptual framework, and then as a proof of concept case study using the medication Praziquantel, a parasitic and antihelminth medication essential in the treatment of neglected tropical disease (NTD) affecting billions of people.

## The Baseline Spectral Fingerprint sorting concept

Below the two elements of the BSF-S process for inferring quality of a medication are defined: First, the BSF for a known high-quality sample of a medication is defined as a standard against which all future samples of questionable provenance are compared. This is subsequently used to define the sorting system. Second, parameters of the ELECTRA TRI-B sorting algorithm are trained to correctly sort factually authentic and factually LQ/C samples.

### The Baseline Spectral Fingerprint (BSF)

A prepared sample of a known factually authentic medication in solution is placed in an UV spectrophotometer. Measures of transmitted light absorbance are captured at discrete wavelengths evenly distributed across on the segment of the UV spectrum across which the active pharmaceutical compound is reactive to UV light. This is conceptually illustrated below for three hypothetical medications called samples (Fig 1). Here, each sample was purposefully prepared such that an absorbance equal to 1.0 at 240 nm is achieved (for purposes explained in the proof of concept section of this paper). These measures of absorbance are defined as the BSF for the medication. For example: Sample "A" has a BSF of [1.0, 0.8, 0.6, 0.4] at wavelengths [240, 250, 260, 270] nm.

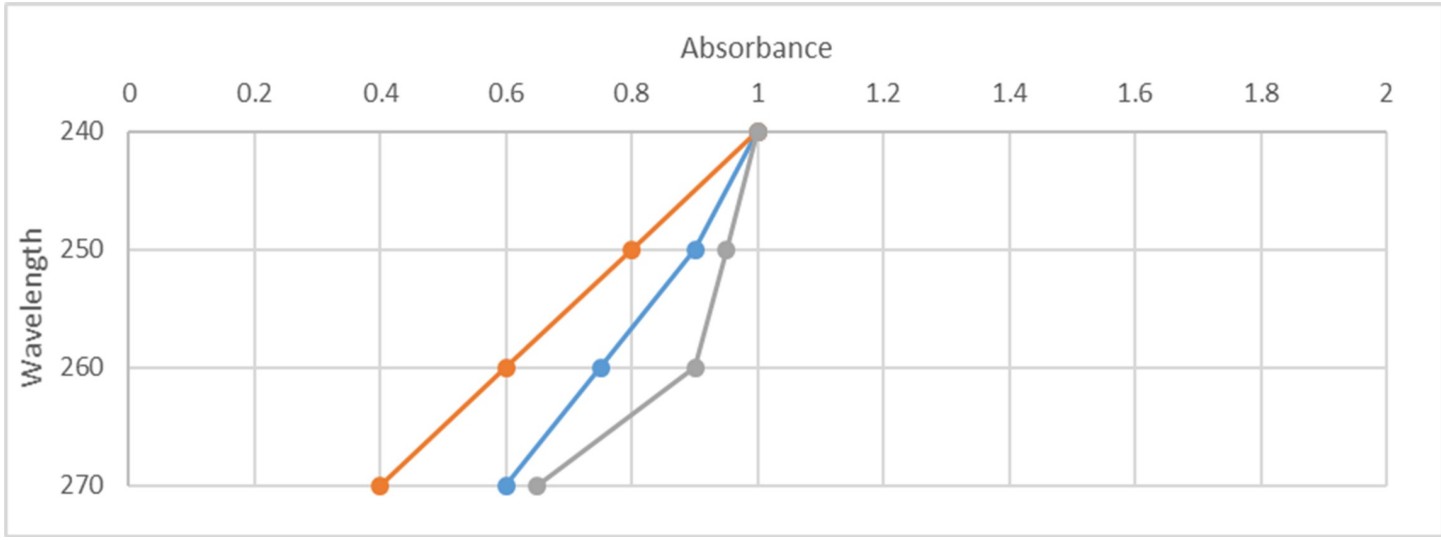

**Fig 1. Conceptual illustration of theoretical BSFs for three medications.** Illustration of three samples with transmitted light absorbance plotted at 240, 250, 260 and 270 nm, with the target baseline dilution of 1.0 for 240 nm. Note the absorbance scale is bounded at zero (transparent solution) and 2 (opaque solution).

## The sorting algorithm (S)

The BSF returns discrete values of absorbance at discrete wavelengths in the UV spectrum. Even under the most ideal testing circumstances, deterministic values as a baseline have limitations due to the high probability that two identical samples (even if prepared at the utmost precision) will not return identical values of absorbance beyond the first digit (for example, an absorbance of 1.0 versus 1.1). This yields to the question of just how far from this baseline a sample can be and still be defined as "authentic" or not. The obvious challenge being the likelihood of false negatives (high quality samples being rejected as LQ/C), and false positives (factually LQ/C samples being defined as authentic).

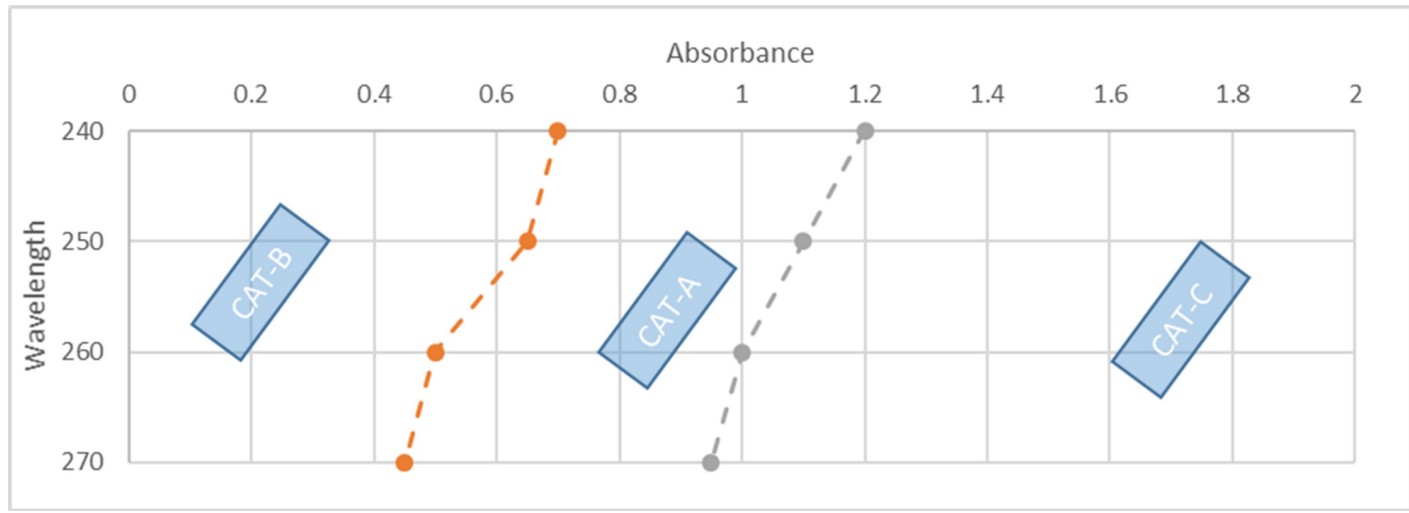

**Fig 2. Conceptual illustration of three categories and two boundaries.** Lower boundary of CAT-B equals zero on absorbance scale (transparent solution). Upper boundary of CAT-B equals lower boundary of CAT-A. Upper boundary of CAT-A equals lower boundary of CAT-C. Upper boundary of CAT-C equals 2 (opaque solution).

To compensate, the BSF data was used as the input to define a category system with upper and lower acceptability boundaries for each wavelength in concert with the ELECTRE TRI-B sorting algorithm. Each sample can then be sorted into authentic or LQ/C categories based on comparative assessment of the sample's absorbance metrics to the BSF. This is conceptually illustrated below in Figs 2–5.

In Fig 2, we illustrate a single central category (CAT-A) into which the BSF-S sorts samples with absorbance profiles similar to the BSF. Note that the system is calibrated such that an absorbance of one is central to the system. This was purposeful so that absorbance readings further into the UV spectrum would not get crowded at the 0 or 2 endpoints of the absorbance scale. The specifics as to the range between lower and upper CAT-A boundaries are detailed in the proof of concept section of this paper. The two adjacent categories (CAT-B & CAT-C) are by default where LQ/C samples are sorted.

To further this conceptual presentation, Fig 3 illustrates a "Sample-X" with resulting measures of absorbance plotted (Fig 3). As graphically illustrated, the sample can easily be visually sorted to CAT-A.

Sorting Sample-X to CAT-A is not controversial. However, we expect challenges to the system when implemented in field conditions, conceptually illustrated in Fig 4, wherein there is less visual clarity as to which category the "Sample-Y" should be sorted (Fig 4). Note that three of the four absorbance readings are within the CAT-A upper and lower boundaries, however, one reading is not, but it is close. In this instance, it is not necessarily as clear whether Sample-Y should be sorted as authentic or LQ/C. And it is for this type of situation that the ELECTRE TRI-B algorithm was employed to aid the sorting problem, particularly in an automated device that avoids visual interpretation of graphic illustrations.

There are a number of algorithms available that sort or cluster items into categories using parameters set by the user. K-Means Clustering and PROMETHEE Sort are two common examples. ELECTRE TRI-B was selected, however, for use in this research because the flexibility and robustness of the parameters that surround category boundaries allowed for a high-resolution analysis of the absorbance data extracted from the sample [17, 18] (S1 Fig) (Fig 5). Specifically, ELECTRE TRI-B uses three uncertainty parameters that surround the boundaries

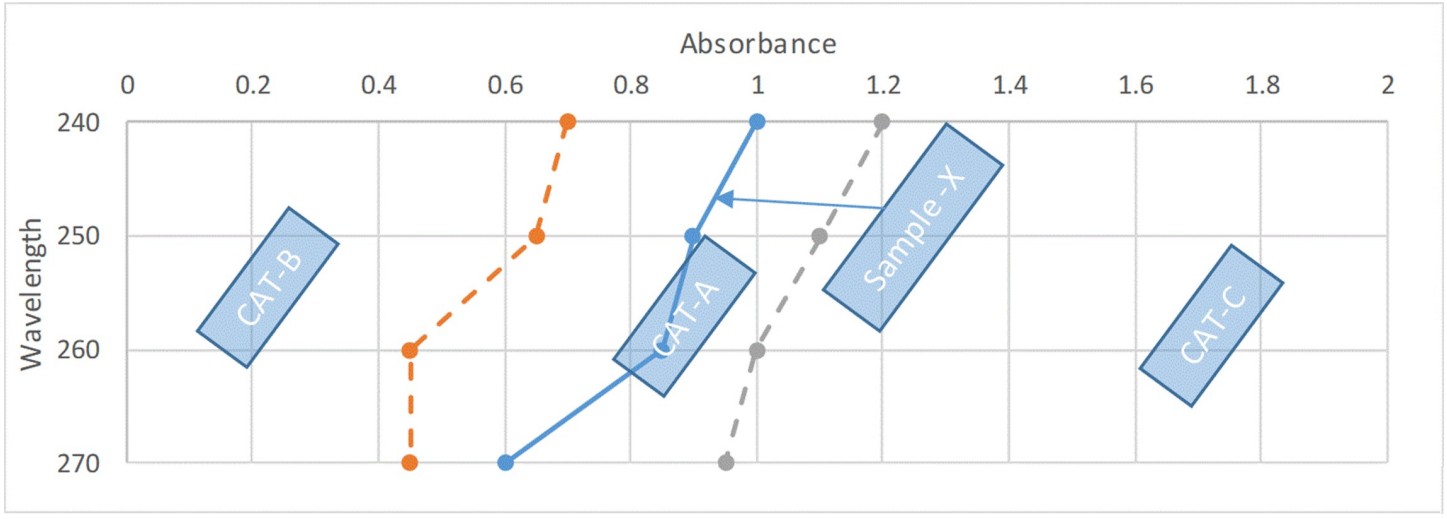

**Fig 3. Conceptual illustration of category system and sample.** CAT-A, CAT-B and CAT-C boundaries and sorting system are illustrated with a hypothetical Sample-X. Here it can be seen that the spectral fingerprint of Sample-X places the sample within the boundaries of CAT-A which would be assumed an authentic sample of the medication.

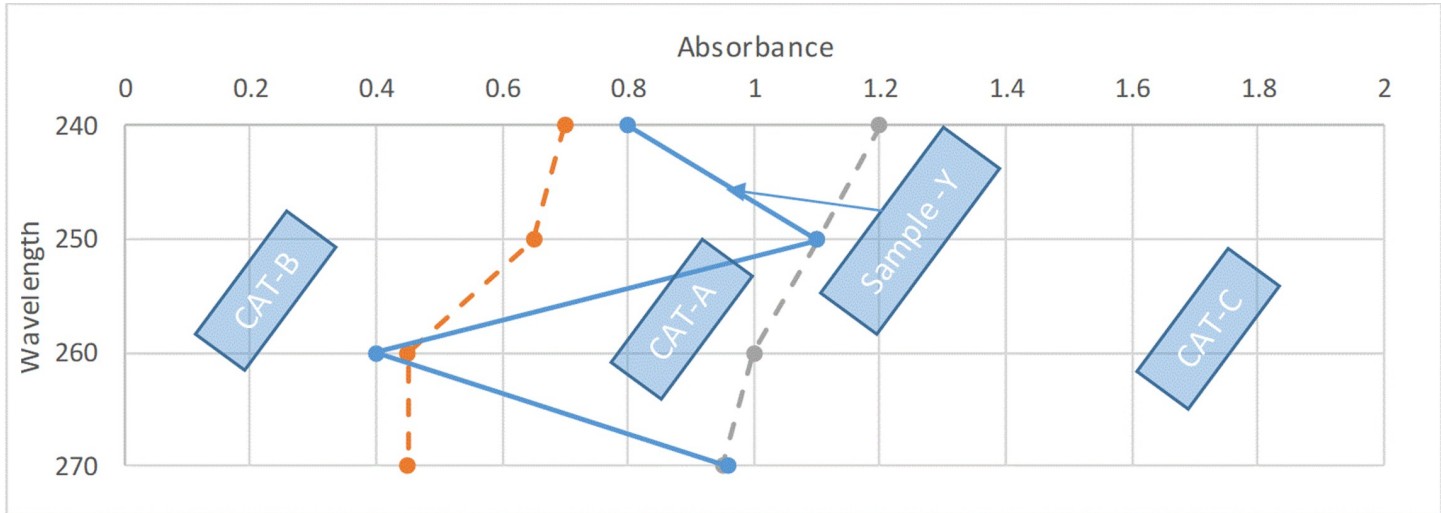

**Fig 4. Conceptual illustration of uncertainty of sorting.** Hypothetical Sample-Y yield ambiguity as to which category sample should be sorted.

for each category in the sorting system: 1) indifference parameter, 2) preference parameter, and 3) veto parameter. These are conceptually illustrated in Fig 5 juxtaposed with the Sample-Y from Fig 4. Here it can be conceptually understood that the uncertainty parameters are used to facilitate proper sorting of these more controversial samples.

The ELECTRE TRI-B process is a paired-comparison of each item against the boundaries (and uncertainty parameters) in a left-to-right sweep, and then a right-to-left sweep in order to build a credibility index that is in effect a statement as to the strength of the argument that an item should be sorted to a specific category. In the context of this research and the sorting system illustrated in Fig 5 with two boundaries (CAT-A lower and upper boundary), the left-to-right sweep first tests the sample's absorbance at each wavelength against the CAT-A lower

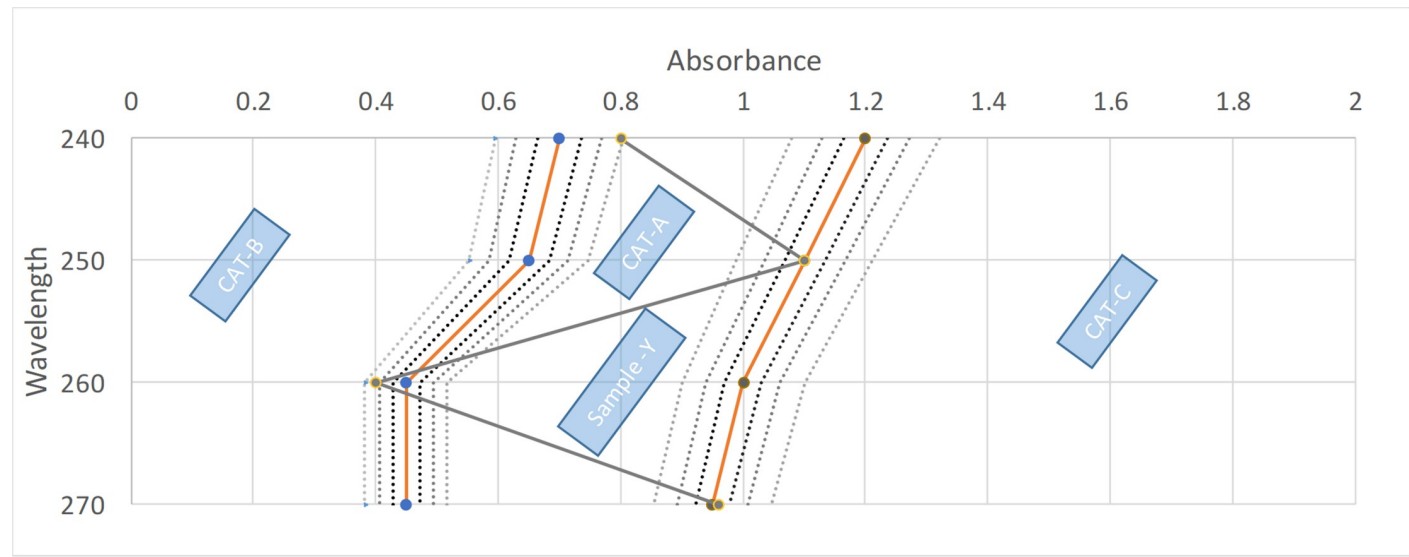

**Fig 5. Conceptual illustration of categories and uncertainty boundaries.** Conceptual illustration of the ELECTRE TRI-B category sorting system with complete set of uncertainty parameters around each CAT-A boundary threshold.

boundary in context of the question "is the sample at least as good as the boundary"? If no, then the sample is preliminarily sorted to CAT-B. If yes, then the procedure is repeated against the upper CAT-A boundary. If the result of this second test is 'no', then the sample is preliminarily sorted to CAT-A, otherwise it is preliminarily sorted to CAT-C.

This initial left-to-right sweep is insufficient to determine a definitive sorting to a specific category because it considers each metric of absorbance independently, which subsequently yields the possibility for false sorting to CAT-A under certain circumstances conceptually illustrated as Fig 6.

Here the hypothetical "Sample Y" has measures of absorbance at 240, 250, 260 and 270 nm such that two (240 and 260 nm) are factually located near the upper boundary of CAT-B. The remainder (250 and 270 nm) are located in CAT-C. Based on the logic of the left-to-right sweep, Sample-Y could be sorted to CAT-A. This is because Sample-Y is at-least-as-good-as the lower upper boundary of CAT-B, but not at-least-as-good-as the upper boundary of CAT-A. This specific situation would result in a false positive, when in fact no element of the absorbance data is physically located in CAT-A

To counter this situation, the ELECTRE TRI-B algorithm requires a right-to-left sweep that repeats the process in reverse to answer the question "is the boundary at least as good as the sample". For the hypothetical Sample-Y of Fig 6, this results in 250 and 270 nm data causing the sample to not be sorted to CAT-A, thus avoiding the false-positive-problem.

In the general application of ELECTRE TRI-B this situation causes Sample-Y to be defined as "incomparable" to the sorting system. In other words, the sample cannot logically be sorted to any category of the sorting system. This characteristic of the ELECTRE TRI-B algorithm is attractive because when sorting medications to authentic and LQ/C categories, any sample defined as incomparable to the sorting system is in effect not an authentic sample, and thus is subsequently defined as LQ/C. This is logically sound and convenient simply because any random substance might be used to create a counterfeit medication, and this logic reduces ambiguity: any sample not explicitly sorted to CAT-A (i.e. sorted to CAT-B, CAT-C or defined as incomparable to the sorting system) is by definition LQ/C.

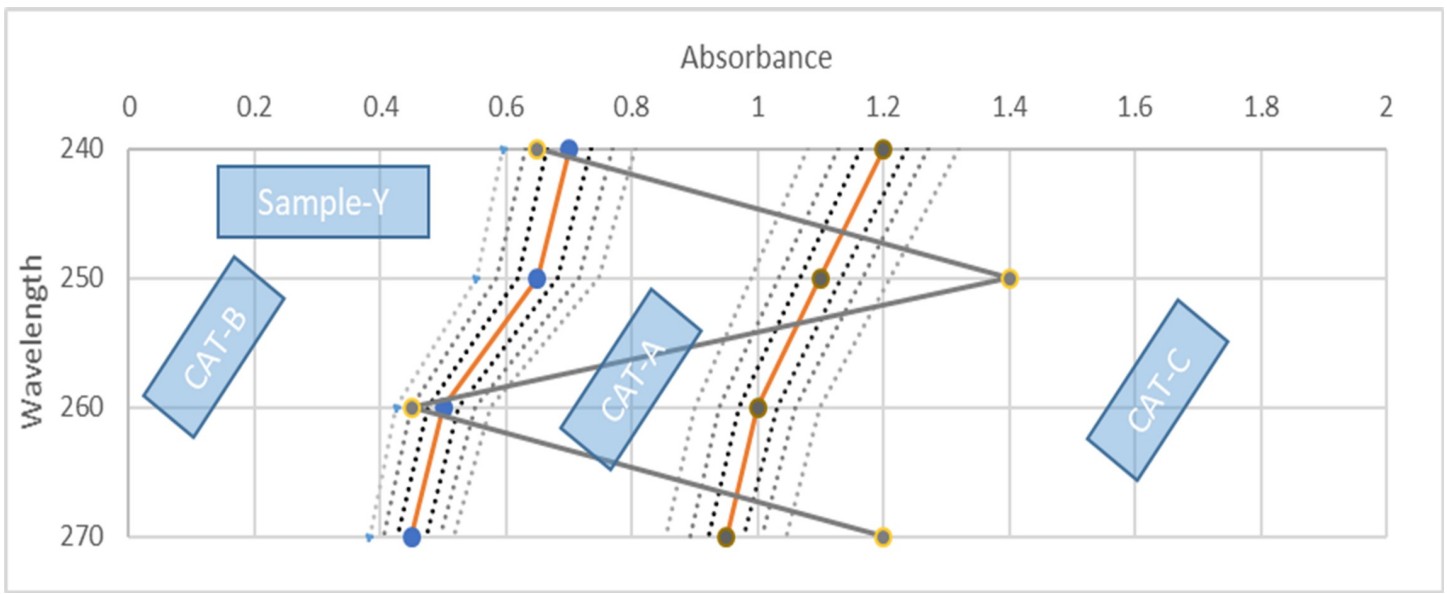

**Fig 6. Example of potential false positive sorting.** Sample-Y is illustrated with four measures of absorbance, one at each of the four wavelengths in the BSF: two are located in CAT-B, two are located in CAT-C.

From this conceptual presentation it can clearly be seen that the ELECTRE-TRI-B uncertainty parameters are critical components to the success of the BSF-S method.

## Proof of concept: Praziquantel

For the specific application presented here, the method to define ELECTRE TRI-B boundary and parameter values is demonstrated as a proof of concept using Praziquantel to test the BSF-S system.

### Baseline Spectral Fingerprint—Praziquantel

High quality samples of Praziquantel were purchased at CVS pharmacy #0447 Northampton MA USA NDC: #50419-0747-01 (Aldrich). The segment of the UV spectrum along which Praziquantel is reactive to UV light was determined to be between 220 to 270 nm using an Agilent Cary UV/VIS spectrophotometer, with the subsequent BSF for Praziquantel defined at wavelengths 240nm, 250nm, 260nm and 270 nm primarily because the technology of UV LED is limited below 240nm. Discussions with suppliers indicated that 240nm LED were a stable and relatively low-cost technology at $400 per unit, with a useful life exceeding 1,000 hours. 230nm and lower into the UV spectrum were, however, experimental with a cost of $1,000 per unit and expected life less than 100 hours.

Samples of high quality Praziquantel were prepared using simplified sample preparation procedures in order to mimic the conditions expected to be used when preparing field samples for testing in LIC/CS., and thus mimic the variability of field prepared samples.

- First, a solvent is selected to dissolve the pill form of the medication into solution using two guidelines: 1) the solvent is more likely than not to be accessible to investigators in LIC/CS countries, and 2) the solvent adequately dissolves the active pharmaceutical compound and excipient of the medication. It is assumed that there is no access to sonicators, costly filtration paper or exotic/costly solvents. Thus, to dissolve the pill form of the medication: 1) the pill is crushed using a mortar and pestle until it is a consistent fine powder, 2) a fixed mass of the power is stirred into a solution using a fixed volume of solvent, 3) the solution is filtered through a gauze pad to remove solids.

- Second, the solution is diluted so that an absorbance of approximately 1.0 (scale 0–2) is achieved at the lowest selected wavelength of the baseline spectrum.

- Third, transmitted light absorbance at the baseline spectrum wavelengths are captured and recorded as the BSF of the medication.

### ELECTRE TRI-B sorting algorithm—Praziquantel

**CAT-A upper and lower boundaries defined.** The BSF process outlined above returns a single realization of the BSF. If the process were repeated, a different realization is highly likely with different values at each wavelength and recorded BSF. This reality was leveraged to define the upper and lower absorbance boundaries for CAT-A in the BSF-S system by repeating the experiment 15 times using 15 individual samples prepared from the procured Praziquantel, each with a unique BSF. As a note, the 15 samples were not intended to be a statistically significant number of samples, but rather a sufficiently large enough sample that the general variability in sample preparation was captured in the process. The captured absorbance data are presented in Fig 7. The lowest and highest absorbance measures from these 15 samples at each

wavelength were determined to be the default lower and upper boundary for the CAT-A, which are illustrated in Fig 7 as connecting lines

The data illustrated in Fig 7 are presented in Table 1 as numeric values for the upper and lower boundaries of CAT-A.

**Uncertainty parameters defined.** The upper and lower boundaries of the sorting system defined above were determined to in effect negate the need for inclusion of the indifference parameter $q_j(b_h)$ in the ELECTRE TRI-B sorting algorithm. Thus, to define the preference and veto parameters, twenty-five samples were prepared by a third-party pharmacist, of which four were authentic Praziquantel, with the remainder being different substances selected to mimic chemical and/or visual properties of Praziquantel, and thus to function as proxy LQ/C samples. The resulting absorbance data are presented in Table 2. These data were used to train the algorithm's preference and veto parameters by minimizing false positives and false negatives. Table 3 contains the resulting parameter values. As a note, Credibility Threshold value was not calibrated using this process, but was set at a user defined 0.85

## Validation

To validate this BSF-S system and defined parameters, 50 samples were prepared by a third-party pharmacist to be have visual or chemical similarity to Praziquantel.. The experimenters were blinded to which samples were authentic and which were proxy LQ/C. It is also noted that some samples were specifically created to mimic LQ (Praziquantel mixed with similar medications) in order to assess some measure of sensitively the BSF-S method. The absorbance data and sorting results are presented in Table 4, which indicate that for the Praziquantel testing regimen, there were no false positives or false negatives, and thus all samples correctly sorted.

A critical note that must be clarified is that: specific to this research and its application of the ELECTRE-Tri-B algorithm any sample tested for which the algorithm yields an

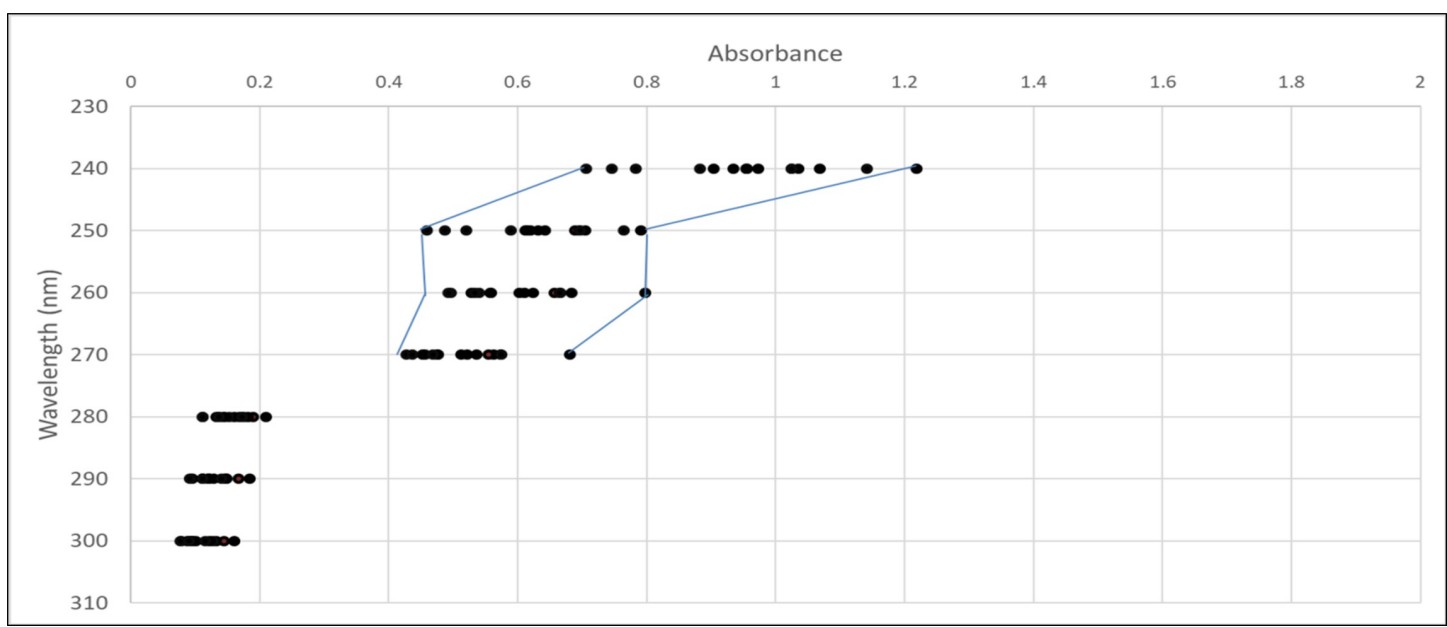

**Fig 7. Absorbance data for 15 samples of authentic Praziquantel prepared using simplified procedures.** Variability in resulting concentration of Praziquantel in solution induced variability in subsequent measures of absorbance.

**Table 1. Lower and upper boundaries for Praziquantel CAT-A.**

| Wavelength | Absorbance | | |
|---|---|---|---|
| | Lower Boundary | Mean | Upper Boundary |
| 240 nm | 0.71 | 0.97 | 1.22 |
| 250 nm | 0.44 | 0.62 | 0.79 |
| 260 nm | 0.46 | 0.63 | 0.80 |
| 270 nm | 0.40 | 0.54 | 0.68 |

These data represent absorbance of 15 independent samples of Praziquantel, sourced from a known supplier. Each sample was prepared using simplified preparation procedures in mimic randomness expected to be found in field operations. Lowest and highest values were assumed to be lower and upper boundary conditions for ELECTRE TRI sorting system.

incomparable result, that sample was defined as LQ/C and reported as correctly sorted in Table 4. Further, ELECTRE TRI-B also allows for an indifferent finding for instances wherein a sample can be sorted to either of two adjacent categories at the decision makers discretion. In this instance, samples were again assumed to be LQ/C, and again defined as correctly sorted. In both cases, incomparable and indifferent, the claim that these are "sorted correctly"

**Table 2. Absorbance data for 25 samples used to optimize uncertainty parameters.**

| | Wavelength | | | | Factual Substance |
|---|---|---|---|---|---|
| Sample | 240 | 250 | 260 | 270 | |
| 1 | 0.471 | 0.91 | 1.779 | 1.485 | Phenylboronic Acid |
| 2 | 0.558 | 0.379 | 0.373 | 0.318 | Azithromycin & Praziquantel |
| 3 | 1.51 | 0.324 | 0.153 | 0.136 | Sodium Nitrate |
| 4 | 0.714 | 0.45 | 0.474 | 0.412 | Praziquantel |
| 5 | 0.15 | 0.099 | 0.083 | 0.077 | Myristic Acid |
| 6 | 0.61 | 0.611 | 0.904 | 0.889 | 4-chlorobenzyl Alcohol |
| 7 | 2 | 2 | 2 | 2 | 4-biphenyl methanol |
| 8 | 0.393 | 0.331 | 0.227 | 0.173 | Sodium Sulfite |
| 9 | 0.322 | 0.246 | 0.163 | 0.135 | Sodium Chloride |
| 10 | 0.339 | 0.252 | 0.164 | 0.142 | Sodium Bicarbonate |
| 11 | 0.776 | 0.532 | 0.439 | 0.365 | Praziquantel & Ivermectin |
| 12 | 0.447 | 0.328 | 0.183 | 0.16 | Magnesium Sulfate |
| 13 | 2 | 1.688 | 2 | 2 | methyl-triphenylphosphonium bromide |
| 14 | 0.862 | 0.565 | 0.522 | 0.448 | Praziquantel |
| 15 | 0.393 | 0.278 | 0.147 | 0.123 | Sodium Sulfate |
| 16 | 0.357 | 0.194 | 0.203 | 0.205 | Sweet n' Low |
| 17 | 0.423 | 0.294 | 0.152 | 0.129 | Sugar |
| 18 | 0.414 | 0.288 | 0.143 | 0.12 | Splenda |
| 19 | 0.887 | 0.566 | 0.516 | 0.439 | Praziquantel |
| 20 | 0.472 | 0.321 | 0.199 | 0.198 | Coffee |
| 21 | 0.434 | 0.287 | 0.128 | 0.104 | Sugar |
| 22 | 0.678 | 0.317 | 0.349 | 0.364 | Sweet n' Low |
| 23 | 0.43 | 0.281 | 0.136 | 0.114 | Splenda |
| 24 | 1.253 | 1.153 | 1.059 | 0.972 | Pepper |
| 25 | 0.907 | 0.584 | 0.56 | 0.478 | Praziquantel |

Raw transmitted light absorbance data collected for 25 samples prepared in order to calibrate ELECTRE TRI uncertainty parameters.

**Table 3. Uncertainty parameters for lower and upper boundaries of CAT-A.**

| | |
|---|---|
| Indifference | 0 |
| Preference | 0.05 |
| Veto | 0.10 |
| Credibility Threshold | 0.85 |

Uncertainty parameters for ELECTRE TRI sorting algorithm are displayed. Values were calibrated to minimize number of false positive and false negatives out of 25 samples of random substances (with 4 being factually positive). Parameters were held as constant across all four criteria.

is a modest and fair adaptation of a strict adherence to the ELECTRE TRI-B process simply because in the specific instance of using the process to determine authenticity of a medication, if a sample cannot be sorted to CAT-A, it is by definition LQ/C.

As stated previously, the credibility threshold ($\lambda$) was user defined and held constant at 0.85 as a reasonable value that requires a substantive amount of evidence that an item should be sorted to a category, but not an overly burdensomely high barrier. To test system sensitivity of the BSF-S system to $\lambda$, the analysis was repeated while varying $\lambda$ between 0 and 1 (inclusive). Table 5 presents results. Here it can be seen that at $\lambda < 0.82$, one of the factually authentic samples (Sample #13 from Table 4) was sorted as LQ/C. Further, as $\lambda$ was increased above 0.98 Sample #13 was again sorted to LQ/C.

## Discussion

In this paper, we describe a BSF-S method used to infer quality of samples of medications being tested with a validation process that tests factually authentic and factually LQ/C samples of Praziquantel as a representative medication that is imperative for the treatment of NTDs affecting millions of people annually. The BSF-S method was developed explicitly for use in LIC/CS, where the capacity for national governments to oversee national drug stock is at its weakest, and the exposure of these populations to LQ/C drugs at its greatest. We conceived and developed this method to add a layer of protection at, or near, the point-of-care such that it is within the technical expertise of local healthcare providers and laboratory technicians, but whom have limited (or no) access to adequate laboratory equipment. Further, the BSF-S method was conceived to be used in conjunction with a UV LED-based spectrophotometer device that is currently under development, and which will be reported at a later date.

In developing the BSF-S method, we optioned to use UV-LED technology, rather than a full-scale spectrum device, with a broad-spectrum bulb, in order to keep costs minimal and maximize the durability and portability of the future associated UV-LED based device. Further, we specifically chose to develop a methodology knowing that when field-deployed there would be limits in associated resources, such as sonicators, advanced solvents and filtration paper. Thus, the resultant method utilizes a simplified sample preparation process, albeit with an expected increase in the variability and randomness in prepared sample concentrations. We compensate for this by integrating the ELECTRE TRI-B sorting algorithm into the analytic device using uncertainty parameters that were calibrated in the laboratory using a mixture of factually authentic and purposefully created LQ/C samples.

Randomness is common in stochastic systems, thus is commonly solved with statistically-based sampling methods. However, as a stochastic/statistical method, there is required a large number of samples in order to satisfy the rules of probability (i.e. satisfy a hypothesis test). In our BSF-S method, we chose instead to 'flip' the analysis requirements so that the large number of samples were required in the laboratory when developing the sorting system and calibrating

**Table 4. Absorbance data for 50 samples used to test BSF data for Praziquantel.**

| Sample | Wavelengths | | | | Factual Substance | Sorted Category | True Category |
|---|---|---|---|---|---|---|---|
| | 240 | 250 | 260 | 270 | | | |
| 1 | 0.776 | 0.532 | 0.439 | 0.365 | Praziquantel & Ivermectin (proxy LQ) | LQ/C | LQ/C |
| 2 | 0.339 | 0.252 | 0.164 | 0.142 | Sodium Bicarbonate (C) | LQ/C | LQ/C |
| 3 | 0.907 | 0.584 | 0.56 | 0.478 | Praziquantel (A) | A | A |
| 4 | 0.393 | 0.278 | 0.147 | 0.123 | Sodium Sulfate (C) | LQ/C | LQ/C |
| 5 | 0.43 | 0.281 | 0.136 | 0.114 | Splenda (C) | LQ/C | LQ/C |
| 6 | 0.322 | 0.246 | 0.163 | 0.135 | Sodium Chloride (C) | LQ/C | LQ/C |
| 7 | 0.862 | 0.565 | 0.522 | 0.448 | Praziquantel (A) | A | A |
| 8 | 0.678 | 0.317 | 0.349 | 0.364 | Sweet n' Low (C) | LQ/C | LQ/C |
| 9 | 0.43 | 0.281 | 0.136 | 0.114 | Splenda (C) | LQ/C | LQ/C |
| 10 | 2 | 2 | 2 | 2 | 4-biphenyl methanol (C) | LQ/C | LQ/C |
| 11 | 0.678 | 0.317 | 0.349 | 0.364 | Sweet n' Low (C) | LQ/C | LQ/C |
| 12 | 1.253 | 1.153 | 1.059 | 0.972 | Pepper (C) | LQ/C | LQ/C |
| 13 | 0.714 | 0.45 | 0.474 | 0.412 | Praziquantel (A) | A | A |
| 14 | 0.393 | 0.278 | 0.147 | 0.123 | Sodium Sulfate (C) | LQ/C | LQ/C |
| 15 | 2 | 1.688 | 2 | 2 | methyl-triphenylphosphonium bromide | LQ/C | LQ/C |
| 16 | 0.322 | 0.246 | 0.163 | 0.135 | Sodium Chloride (C) | LQ/C | LQ/C |
| 17 | 0.357 | 0.194 | 0.203 | 0.205 | Sweet n' Low (C) | LQ/C | LQ/C |
| 18 | 0.887 | 0.566 | 0.516 | 0.439 | Praziquantel (A) | A | A |
| 19 | 0.434 | 0.287 | 0.128 | 0.104 | Sugar (C) | LQ/C | LQ/C |
| 20 | 0.471 | 0.91 | 1.779 | 1.485 | Phenylboronic Acid (C) | LQ/C | LQ/C |
| 21 | 0.423 | 0.294 | 0.152 | 0.129 | Sugar (C) | LQ/C | LQ/C |
| 22 | 1.12 | 0.838 | 0.272 | 0.135 | Ivermectin (C) | LQ/C | LQ/C |
| 23 | 0.888 | 0.667 | 0.238 | 0.132 | Ivermectin (C) | LQ/C | LQ/C |
| 24 | 1.51 | 0.324 | 0.153 | 0.136 | Sodium Nitrate (C) | LQ/C | LQ/C |
| 25 | 0.15 | 0.099 | 0.083 | 0.077 | Myristic Acid (C) | LQ/C | LQ/C |
| 26 | 0.15 | 0.099 | 0.083 | 0.077 | Myristic Acid (C) | LQ/C | LQ/C |
| 27 | 0.61 | 0.611 | 0.904 | 0.889 | 4-chlorobenzyl Alcohol (C) | LQ/C | LQ/C |
| 28 | 0.447 | 0.328 | 0.183 | 0.16 | Magnesium Sulfate (C) | LQ/C | LQ/C |
| 29 | 0.423 | 0.294 | 0.152 | 0.129 | Sugar (C) | LQ/C | LQ/C |
| 30 | 0.472 | 0.321 | 0.199 | 0.198 | Coffee (C) | LQ/C | LQ/C |
| 31 | 0.414 | 0.288 | 0.143 | 0.12 | Splenda (C) | LQ/C | LQ/C |
| 32 | 0.357 | 0.194 | 0.203 | 0.205 | Sweet n' Low (C) | LQ/C | LQ/C |
| 33 | 1.245 | 0.921 | 0.273 | 0.117 | Ivermectin (C) | LQ/C | LQ/C |
| 34 | 1.253 | 1.153 | 1.059 | 0.972 | Pepper (C) | LQ/C | LQ/C |
| 35 | 2 | 1.688 | 2 | 2 | methyl-triphenylphosphonium bromide | LQ/C | LQ/C |
| 36 | 1.51 | 0.324 | 0.153 | 0.136 | Sodium Nitrate (C) | LQ/C | LQ/C |
| 37 | 0.471 | 0.91 | 1.779 | 1.485 | Phenylboronic Acid (C) | LQ/C | LQ/C |
| 38 | 0.414 | 0.288 | 0.143 | 0.12 | Splenda (C) | LQ/C | LQ/C |
| 39 | 1.68 | 1.839 | 1.282 | 0.747 | Ivermectin (C) | LQ/C | LQ/C |
| 40 | 2 | 2 | 2 | 2 | 4-biphenyl methanol (C) | LQ/C | LQ/C |
| 41 | 0.393 | 0.331 | 0.227 | 0.173 | Sodium Sulfite (C) | LQ/C | LQ/C |
| 42 | 0.393 | 0.331 | 0.227 | 0.173 | Sodium Sulfite (C) | LQ/C | LQ/C |
| 43 | 0.434 | 0.287 | 0.128 | 0.104 | Sugar (C) | LQ/C | LQ/C |
| 44 | 0.472 | 0.321 | 0.199 | 0.198 | Coffee (C) | LQ/C | LQ/C |
| 45 | 0.558 | 0.379 | 0.373 | 0.318 | Azithromycin & Praziquantel (Proxy LQ) | LQ/C | LQ/C |
| 46 | 0.776 | 0.532 | 0.439 | 0.365 | Praziquantel & Ivermectin (proxy LQ) | LQ/C | LQ/C |

(*Continued*)

**Table 4.** (Continued)

| Sample | Wavelengths | | | | Factual Substance | Sorted Category | True Category |
|---|---|---|---|---|---|---|---|
| | 240 | 250 | 260 | 270 | | | |
| 47 | 0.61 | 0.611 | 0.904 | 0.889 | 4-chlorobenzyl Alcohol (C) | LQ/C | LQ/C |
| 48 | 0.339 | 0.252 | 0.164 | 0.142 | Sodium Bicarbonate (C) | LQ/C | LQ/C |
| 49 | 1.101 | 0.829 | 0.259 | 0.124 | Ivermectin (C) | LQ/C | LQ/C |
| 50 | 0.447 | 0.328 | 0.183 | 0.16 | Magnesium Sulfate (C) | LQ/C | LQ/C |

Results of BSF-S regiment for 50 samples of factually authentic, proxy LQ and factually C samples. All items sorted to either CAT-B, CAT-C or are defined as Indifferent or Incompatible are listed as LQ/C

parameters for any one specific drug. For example, if the BSF-S were determined to a high resolution in the lab using highly sophisticated equipment, then simple point-values data collected from samples in the field would not be sufficient to infer quality because any deviation from the high-resolution lab results would have to be assumed as LQ/C.

To mitigate this challenge, we incorporated the ELECTRE TRI-B sorting algorithm in the laboratory to define the boundary conditions and uncertainty parameters with which a single sample prepared in the field would suffice to infer quality. We believe that mimicking field conditions in the development laboratory mitigated recognizable limiting factors by embracing these factors early in the design process. To our knowledge, this is a novel approach as sorting approaches have not been used previously to ascertain medication authenticity in any setting.

There are a number of sorting algorithms that consider uncertainty in data. However, our preliminary research found that the nuances of the ELECTRE TRI-B were superior for the purposes of sorting medication into authentic and LQ/C categories, specifically because of the veto parameter that is user defined. The veto threshold sets the credibility index that a sample belongs to the authentic category to zero if any single measure of absorbance exceeds the defined veto threshold. This is a powerful characteristic because without the veto threshold, any random substance that scores high in three out of four measures, but scores poorly in just one, might still be sorted to the authentic category. The veto threshold avoids this problem by defining a sample as incomparable to a boundary, and thus in our application is automatically defined as LQ/C.

In the method and data presented in this paper, we limited the category choice to authentic and LQ/C. However, the designation LQ/C was intended to capture counterfeit as well as low quality. Further, the proof of concept study presented included enough data to allow the exploration of applicability of the method to detect LQ separately from C. Specifically, Samples 1, 33 and 46 (see Table 4) were either authentic samples of Praziquantel mixed with other authentic medication of similar chemistry (Ivermectin and Azithromycin), or one of these other

**Table 5. Credibility threshold sensitivity analysis.**

| | Credibility Threshold (λ) | | |
|---|---|---|---|
| | < .82 | 0.82–0.98 | >0.98 |
| False Negative | 1/4 | 0/50 | 1/4 |
| False Positive | 0/50 | 0/50 | 0/50 |

Data illustrates no false positives for any value of λ, with one sample being falsely sorted as LQ/C for values below 0.82 and above 0.98

medications. The BSF-S method classified these samples as 'indifferent' to either the lower or upper boundary of CAT-A. For this initial research, we interpreted this result as LQ/C.However, given that these samples were purposefully prepared to test the resolution of the BSF-S method, this is a clear indicator that there is potential to include a separate LQ category located between CAT-B & CAT-A, and between CAT-A & CAT-C. Knowing that a sample is LQ is interesting for quality control and root cause analysis purposes by the in-field investigator, and we will revisit this potential as the companion UV-LED-based device research advances.

The BSF-S as presented is relatively sensitive to the Credibility Threshold value ($\lambda$). Table 5 illustrates that deviation to 0.82 would result in one authentic sample being sorted as LQ/C. This challenge itself can be easily corrected by lowering the CAT-A lower boundary (and raising the upper boundary). While it may seem to be arbitrary to increase the width of CAT-A, there is precedence in defining an "intrinsic" lower and upper boundary from a set of stochastic data. Specifically, Cooke uses a value of 10% lower than the lowest value as the lower boundary, and 10% greater than the highest value [19]. A preliminary re-analysis of the data demonstrates that this improves the sensitivity of the BSF-S to values of $\lambda$. Further research will explore including as this type of intrinsic boundary values as an additional parameter to be calibrated in the BSF-S method.

## Conclusion

While not intended to replace current methods of ascertaining medication authenticity in LIC/CS settings, such as visual inspection, supply chain monitoring or PAD devices, the BSF-S method is a viable adjunct/alternative assay method for monitoring regional drug stock quality as an added layer of detection near the point-of-care, and in the hands of local pharmacies and health care providers when national systems are limited.

This paper demonstrates that the BSF-S method is highly sensitive and specific When coupled with a portable UV-LED-based device, there is the potential to add a layer of protection to mitigate human and economic costs associated with LQ/C drugs.

## Supporting information

**S1 File.**
(DOCX)

## Author Contributions

**Conceptualization:** Christian Salmon, Margaret Salmon, Elaine Xu.

**Data curation:** Christian Salmon, Margaret Salmon, Marcus Paoletti, Ronny Priefer, Michael Rust.

**Formal analysis:** Christian Salmon, Elaine Xu, Ronny Priefer, Michael Rust, Aliea Afnan.

**Investigation:** Christian Salmon, Ronny Priefer, Michael Rust, Aliea Afnan.

**Methodology:** Christian Salmon, Ronny Priefer, Michael Rust, Aliea Afnan.

**Software:** Elaine Xu.

**Validation:** Christian Salmon, Aliea Afnan.

**Visualization:** Christian Salmon, Elaine Xu, Michael Rust.

**Writing – original draft:** Christian Salmon, Margaret Salmon, Marcus Paoletti, Elaine Xu, Ronny Priefer, Michael Rust.

**Writing – review & editing:** Christian Salmon, Margaret Salmon, Marcus Paoletti, Elaine Xu, Ronny Priefer, Michael Rust.

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
