## [Decision Letter · Decision Letter 0]

27 Jul 2020

PONE-D-20-14126

A Multi-Criteria Sorting algorithm based method to infer authenticity of medications

PLOS ONE

Dear Dr. Salmon,

Thank you for submitting your manuscript to PLOS ONE. After careful consideration, we feel that it has merit but does not fully meet PLOS ONE’s publication criteria as it currently stands. Therefore, we invite you to submit a revised version of the manuscript that addresses the points raised during the review process.

We look forward to receiving your revised manuscript.

Kind regards,

Fausto Cavallaro, PhD

Academic Editor

PLOS ONE

Journal Requirements:

"the authors received no specific funding for this work"

We note that one or more of the authors are employed by a commercial company: Daymark.

2.1. Please provide an amended Funding Statement declaring this commercial affiliation, as well as a statement regarding the Role of Funders in your study. If the funding organization did not play a role in the study design, data collection and analysis, decision to publish, or preparation of the manuscript and only provided financial support in the form of authors' salaries and/or research materials, please review your statements relating to the author contributions, and ensure you have specifically and accurately indicated the role(s) that these authors had in your study. You can update author roles in the Author Contributions section of the online submission form.

2.2. Please also provide an updated Competing Interests Statement declaring this commercial affiliation along with any other relevant declarations relating to employment, consultancy, patents, products in development, or marketed products, etc. 

Reviewers' comments:

Reviewer's Responses to Questions

**Comments to the Author**

1. Is the manuscript technically sound, and do the data support the conclusions?

Reviewer #1: Yes

Reviewer #2: Yes

2. Has the statistical analysis been performed appropriately and rigorously? 

Reviewer #1: Yes

Reviewer #2: N/A

3. Have the authors made all data underlying the findings in their manuscript fully available?

Reviewer #1: No

Reviewer #2: Yes

4. Is the manuscript presented in an intelligible fashion and written in standard English?

Reviewer #1: Yes

Reviewer #2: Yes

5. Review Comments to the Author

Reviewer #1: Very relevant use of MCDA approach for a societal problem. However, the introduction of the problem has to more clearly done, by giving the required definitions for understanding the ideas. It will be also the case for the choice of ELECTRE-Tri and the highlighting of the used methodology. References should be enriched. Figures and overall schemes would be appreciated for better getting the point of this approach. Please take time also for talking about the MCDA philosophy and its potential application in this sector.

Reviewer #2: Review on PONE-D-20-14126

This paper provides an interesting application of multiple criteria decision aiding (MCDA) model used to infer the authenticity of medications. From a methodological point of view, this is an interesting and well-founded study. The use of ELECTRE and MCDA as a supporting methodological concept is appropriate and well explained. However, several serious concerns need to be addressed. They are mostly related to the structure of the manuscript, precision of the message, and clearness of the whole method presented in the study.

MAJOR REMARKS

1. The structure of the paper is troublesome. In the section “Methods” where you introduce us to your model and the idea behind two components, you are heavily relying on the results of the study we are yet to be introduced with. This paper needs to be significantly restructured. Either you introduce us with the methodology in a formal way and then start with your results, or you create some other smart way. This structure is not a valid one for a reader to fully understand both the method and the idea of your proposal and corresponding results.

2. There are several methods within the ELECTRE family of methods. Claiming that you use ELECTRE TRI is not clear enough. The first one, at the start, was called just ELECTRE TRI (Yu, 1992). Later on, after the development of other methods, it was named ELECTRE Tri-B. Its characteristic is that the categories in the method are defined via boundary profiles set by decision-maker (DM). The newest method within the family is ELECTRE Tri-nB (Fernández et al., 2017). The “n” in the name suggests that DM is now allowed to set any number of boundary profiles between two categories. On the other hand, ELECTRE Tri-C (Almeida-Dias et al., 2010) introduced a novel way of thinking by setting, not boundary profiles, but characteristic profile, factoring as the most representative alternative for each category. In a similar fashion ELECTRE Tri-nC is a method where you can have any number of characteristic profiles.

Reading the paper, it becomes clear which model you use, but it is necessary to be precise. Moreover, the reason for my comment is the complete omission of the references at the moment when you introduce the method. More on reference later. Also, as ELECTRE Tri-B is a doctoral dissertation written in French, using reference (12) at that point is justified.

3. I have a major problem with the organization and the names of the subsection in the “Methods” section. Are definitions necessary in the manner you introduced them? It is very uncommon, and it goes in line with a very strange structure of the paper. Moreover, you put heading as if it is the section of the same level. It is not needed as a subsection. Introduce them in brackets after the first mention like you already at that point did with the LQ/C, for example. No need to repeat yourselves.

Also, naming the next subsection “Introduction to Developing Components 1,2 and Validation” without explanation of the concept is not advised, to say at least (apart from the brief mention in the introduction, but here you have to be much more precise).

4. Line 131. Validation of the BSF-S method required 50 samples… Why did it “require” 50 samples precisely?

5. The references in the paper are even strange and some moments. In some parts, there is a clear void where it should be supported with apparent references. The most striking example is with the notion of ELECTRE TRI, as I already mentioned. Also, when it comes to the elaboration of the ELECTRE and its (completely justified) use in your research, it should be supported with the references of applications based on the ELECTRE family of methods. Preferably in similar studies in the field, but in general as well. You can always check on most prominent journals, check the last two years of publications in top-rated journals in the OR/MS field, and you will find numerous applications of MCDA models in sorting.

If you find a significant number of applications in the field, it speaks in favor of your concept.

If you do not find many applications in your field (either in medical or OR journals), it would be beneficial to elaborate on such a finding. Why is that the case? Was there a reason to opt for some other methods? Is that a gap that could be beneficial? Could this yield several novel applications in the field of the known problems? As you see, this can be a very valid point for the deeper elaboration of your concept and contribution.

6. Even though you provide “conceptual” illustrations, proper definitions would be needed. And again, in figures, you use concepts from the study and your results. Even though I understand your desire to explain the model as clearly as possible with these conceptual illustrations, I had big trouble in understanding some parts (Figure 4 as a clear favorite).

A side note is that both pictures and figures are not of good quality, you should try to make it with higher quality standards.

7. Line 205. You name subsection “Part 1 – CAT-A”. Part 1 of what? These are very uncommon terms, let alone to be used as subsections. This goes in line with comments 1 and 3.

8. References in the paper are not according to the submission guidelines and, at some moments, not accurate. A good example is a reference (12) I already mentioned. First of all, the author is not the only editor of that handbook. Stating the date when you checked the book is not common, and it is not according to instructions. Authors need to follow guidelines to the very end and be very precise. This is not a minor mistake at all.

9. I was very interested in the results presented in tables 2 and 3. I’ll state this example to explain my interest here. Was (Would) the conclusion of the decision-maker (and make clearer how did you derive table 3), for a wavelength 240, that there exists a difference in samples 17 and 18 with the results 0.423 and 0.414?

MINOR REMARKS

1. Citations used at the end of the sentences should be before dots. For example, line 47 or at line 50.

2. It is a common practice to introduce the readers at the end of the introduction about the structure of the remainder of the paper (another paragraph after a paragraph ending at line 90).

3. Line 184. Your sentence starts “ELECTRE was chosen as has an”

4. Line 240. Figure 5, instead of Figure 4.

5. Line 302. Dot is missing at the end.

CONCLUSION

The idea of the manuscript is sound. It has the potential to pinpoint some relevant results as well as give a path for future applications of this methodological concept in the field. However, the manuscript is far away from publishable. It is difficult to understand it in some parts. They talk about the results in the section about the method where they use the data from the tables that are yet to be defined. References worryingly differ from the guidelines for authors. Such small problems are present throughout the manuscript. This manuscript has to be heavily restructured and revised in order to be ready for publication.

References

[1] Yu, W. (1992). Aide multicritère à la décision dans le cadre de la problématique du tri: concepts, méthodes et applications, LAMSADE, Université Paris Dauphine, Paris (Ph.D. thesis)

[2] Fernández, E., Figueira, J. R., Navarro, J., & Roy, B. (2017). ELECTRE TRI-nB: A new multiple criteria ordinal classification method. European Journal of Operational Research, 263(1), 214-224.

[3] Almeida-Dias, J., Figueira, J. R., & Roy, B. (2010). ELECTRE Tri-C: A multiple criteria sorting method based on characteristic reference actions. European Journal of Operational Research, 204(3), 565-580.

[4] Almeida-Dias, J., Figueira, J. R., & Roy, B. (2012). A multiple criteria sorting method where each category is characterized by several reference actions: The Electre Tri-nC method. European Journal of Operational Research, 217(3), 567-579.

6. PLOS authors have the option to publish the peer review history of their article (what does this mean?). If published, this will include your full peer review and any attached files.

Reviewer #1: **Yes: **Lamia Berrah

Reviewer #2: No

---

## [Author Response · Author response to Decision Letter 0]

5 May 2021

RESPONSE TO THE REVIEWERS (This is also submitted as an individual document)

FROM. THE EDITOR

 Formatting. - all areas of the manuscript have been checked and they meet the PLOS ONE style requirements

Financial Disclosure - completed

5. Review Comments to the Author  Please use the space provided to explain your answers to the questions above. You may also include additional comments for the author, including concerns about dual publication, research ethics, or publication ethics. (Please upload your review as an attachment if it exceeds 20,000 characters)

Reviewer #1: Very relevant use of MCDA approach for a societal problem. However, the introduction of the problem has to more clearly done, by giving the required definitions for understanding the ideas. It will be also the case for the choice of ELECTRE-Tri and the highlighting of the used methodology. References should be enriched. Figures and overall schemes would be appreciated for better getting the point of this approach. Please take time also for talking about the MCDA philosophy and its potential application in this sector

 After the Introduction, we have added a section called “Baseline Spectral Fingerprint - Sorting” in which we introduce the ideas of both Baseline Spectral Fingerprint and the Multi criteria sorting system. We worked to address the philosophy and applications as requested by the reviewer While we did address the ELECTRE TRI method and identified it as the ELECTRE TRI-C, we do hesitate to increase this section disproportionately at this time. This manuscript was intended for a broad group within the sectors of global health, public health, pharmaceutics and manufacturing as well as engineering.

Line 189 - Line 340

Reviewer #2: Review on PONE-D-20-14126  This paper provides an interesting application of multiple criteria decision aiding (MCDA) model used to infer the authenticity of medications. From a methodological point of view, this is an interesting and well-founded study. The use of ELECTRE and MCDA as a supporting methodological concept is appropriate and well explained. However, several serious concerns need to be addressed. They are mostly related to the structure of the manuscript, precision of the message, and clearness of the whole method presented in the study.

 We agree with the reviewer and the manuscript has been restructured. 

 MAJOR REMARKS  1. The structure of the paper is troublesome. In the section “Methods” where you introduce us to your model and the idea behind two components, you are heavily relying on the results of the study we are yet to be introduced with. This paper needs to be significantly restructured. Either you introduce us with the methodology in a formal way and then start with your results, or you create some other smart way. This structure is not a valid one for a reader to fully understand both the method and the idea of your proposal and corresponding results.

 The. Manuscript has been restructured to better introduce the subject and allow the reader an introduction to the concept then read about its application with Praziquantel. The “Methods” has been replaced with “Proof of Concept.” Please see below. Line 342

 Introduction

 Background

 A new novel testing method 

 The Concept - Baseline Spectral Fingerprint- Sorting 

 Baseline Spectrum Fingerprint

 MultiCriteria Sorting Algorithym

 Proof of Concept - Praziquantel

 Proof of Concept 

 Validation of concept

 Discussion

 Conclusion

 References

 Supplementary Materials

   2. There are several methods within the ELECTRE family of methods. Claiming that you use ELECTRE TRI is not clear enough. The first one, at the start, was called just ELECTRE TRI (Yu, 1992). Later on, after the development of other methods, it was named ELECTRE Tri-B. Its characteristic is that the categories in the method are defined via boundary profiles set by decision-maker (DM). The newest method within the family is ELECTRE Tri-nB (Fernández et al., 2017). The “n” in the name suggests that DM is now allowed to set any number of boundary profiles between two categories. On the other hand, ELECTRE Tri-C (Almeida-Dias et al., 2010) introduced a novel way of thinking by setting, not boundary profiles, but characteristic profile, factoring as the most representative alternative for each category. In a similar fashion ELECTRE Tri-nC is a method where you can have any number of characteristic profiles.  Reading the paper, it becomes clear which model you use, but it is necessary to be precise. Moreover, the reason for my comment is the complete omission of the references at the moment when you introduce the method. More on reference later. Also, as ELECTRE Tri-B is a doctoral dissertation written in French, using reference (12) at that point is justified

 The method, we chose to use is the. ELECtRE-TRI C. This is now clearly identified and referenced. 

 Line 239 - 250

  3. I have a major problem with the organization and the names of the subsection in the “Methods” section. Are definitions necessary in the manner you introduced them? It is very uncommon, and it goes in line with a very strange structure of the paper. Moreover, you put heading as if it is the section of the same level. It is not needed as a subsection. Introduce them in brackets after the first mention like you already at that point did with the LQ/C, for example. No need to repeat yourselves.

 We have restructured the manuscript as described above. We have removed the repetitive sections and also removed the definitions. Definitions had originally been included as this manuscript its intentionally not solely for engineers but rather a broader audience. Definitions are quite common in the literature of global health and International Non Governmental Organizations. We agree, “Definitions” may be distracting at this time.  Also, naming the next subsection “Introduction to Developing Components 1,2 and Validation” without explanation of the concept is not advised, to say at least (apart from the brief mention in the introduction, but here you have to be much more precise).

 This subsection has been retitled. Line 225 is the start  4. Line 131. Validation of the BSF-S method required 50 samples… Why did it “require” 50 samples precisely?

 The sample size was not validated and was chosen by the pharmacist. We have removed any descriptors that would designate it as "required" or “calculated”. Line 464  5. The references in the paper are even strange and some moments. In some parts, there is a clear void where it should be supported with apparent references. The most striking example is with the notion of ELECTRE TRI, as I already mentioned. Also, when it comes to the elaboration of the ELECTRE and its (completely justified) use in your research, it should be supported with the references of applications based on the ELECTRE family of methods. Preferably in similar studies in the field, but in general as well. You can always check on most prominent journals, check the last two years of publications in top-rated journals in the OR/MS field, and you will find numerous applications of MCDA models in sorting.  If you find a significant number of applications in the field, it speaks in favor of your concept.  If you do not find many applications in your field (either in medical or OR journals), it would be beneficial to elaborate on such a finding. Why is that the case? Was there a reason to opt for some other methods? Is that a gap that could be beneficial? Could this yield several novel applications in the field of the known problems? As you see, this can be a very valid point for the deeper elaboration of your concept and contribution. 

 References have been added. Line 245

 6. Even though you provide “conceptual” illustrations, proper definitions would be needed. And again, in figures, you use concepts from the study and your results. Even though I understand your desire to explain the model as clearly as possible with these conceptual illustrations, I had big trouble in understanding some parts (Figure 4 as a clear favorite).

   A side note is that both pictures and figures are not of good quality, you should try to make it with higher quality standards.  7. Line 205. You name subsection “Part 1 – CAT-A”. Part 1 of what? These are very uncommon terms, let alone to be used as subsections. This goes in line with comments 1 and 3.

 The subtitles you refer to have all been removed

  8. References in the paper are not according to the submission guidelines and, at some moments, not accurate. A good example is a reference (12) I already mentioned. First of all, the author is not the only editor of that handbook. Stating the date when you checked the book is not common, and it is not according to instructions. Authors need to follow guidelines to the very end and be very precise. This is not a minor mistake at all.

 The. references. have been rechecked  9. I was very interested in the results presented in tables 2 and 3. I’ll state this example to explain my interest here. Was (Would) the conclusion of the decision-maker (and make clearer how did you derive table 3), for a wavelength 240, that there exists a difference in samples 17 and 18 with the results 0.423 and 0.414?

 Specific to the comment related to 240 nm for samples 17 and 18: the factual substances are Splenda and Sugar, which are chemically similar, and in fact Splenda is designed as a substitute for sugar, and this is demonstrated in the similar readings of absorbance at not just 240, but also 250, 260 and 270 nm: For Sugar [0.423, 0.294, 0.152, 0.129], and for Splenda [0.414, 0.288, 0.143, 0.120].  

 Specific to the Samples 17 &. 18m These two samples, as well as others in Table 2, were selected by a third party pharmacist for the purposes of calibrating the sorting algorithm parameters, which are presented in Table 3, and I believe are relevant to the second part of the reviewer's question: The values in Table 3 were derived by calibrating the parameter values of the algorithm (indifference, preference, veto and credibility threshold) towards minimizing the number of false positives and false negatives.  In order to ensure that the parameters were not overly broad by including chemical substances that only wildly dissimilar absorbance profiles than Praziquantel, proxy low quality samples were included (note Samples 2 and 11) in order to narrow the width of the absorbance pathway to authentic Praziquantel.  The calibration was performed using the Microsoft Excel (2016) solver function, then rounded for convenience.  Table 3 values would need to be redeveloped for each drug for which the parameters are being developed.  If, for example, Albendazole were the subject, there is no expectation that the parameter values would be the same as in Table 3.

   MINOR REMARKS  1. Citations used at the end of the sentences should be before dots. For example, line 47 or at line 50.

 These have been corrected

 2. It is a common practice to introduce the readers at the end of the introduction about the structure of the remainder of the paper (another paragraph after a paragraph ending at line 90).

 This introduction to the structure has been added. Line 169 - 176

 3. Line 184. Your sentence starts “ELECTRE was chosen as has an”

 This has been changed

 4. Line 240. Figure 5, instead of Figure 4.

 References to figures have been corrected

 5. Line 302. Dot is missing at the end.

 Corrected 

CONCLUSION  The idea of the manuscript is sound. It has the potential to pinpoint some relevant results as well as give a path for future applications of this methodological concept in the field. However, the manuscript is far away from publishable. It is difficult to understand it in some parts. They talk about the results in the section about the method where they use the data from the tables that are yet to be defined. References worryingly differ from the guidelines for authors. Such small problems are present throughout the manuscript. This manuscript has to be heavily restructured and revised in order to be ready for publication.  References  [1] Yu, W. (1992). Aide multicritère à la décision dans le cadre de la problématique du tri: concepts, méthodes et applications, LAMSADE, Université Paris Dauphine, Paris (Ph.D. thesis) [2] Fernández, E., Figueira, J. R., Navarro, J., & Roy, B. (2017). ELECTRE TRI-nB: A new multiple criteria ordinal classification method. European Journal of Operational Research, 263(1), 214-224. [3] Almeida-Dias, J., Figueira, J. R., & Roy, B. (2010). ELECTRE Tri-C: A multiple criteria sorting method based on characteristic reference actions. European Journal of Operational Research, 204(3), 565-580. [4] Almeida-Dias, J., Figueira, J. R., & Roy, B. (2012). A multiple criteria sorting method where each category is characterized by several reference actions: The Electre Tri-nC method. European Journal of Operational Research, 217(3), 567-579.

6. PLOS authors have the option to publish the peer review history of their article (what does this mean?). If published, this will include your full peer review and any attached files.   Do you want your identity to be public for this peer review? For information about this choice, including consent withdrawal, please see our Privacy Policy.

Reviewer #1: Yes: Lamia Berrah

Reviewer #2: No

 While revising your submission, please upload your figure files to the Preflight Analysis and Conversion Engine (PACE) digital diagnostic tool, https://pacev2.apexcovantage.com/. PACE helps ensure that figures meet PLOS requirements. To use PACE, you must first register as a user. Registration is free. Then, login and navigate to the UPLOAD tab, where you will find detailed instructions on how to use the tool. If you encounter any issues or have any questions when using PACE, please email PLOS at figures@plos.org. Please note that Supporting Information files do not need this step.

---

## [Decision Letter · Decision Letter 1]

13 Jul 2021

PONE-D-20-14126R1

Ultraviolet fingerprints and a sorting algorithm:  A portable way to authenticate medications

PLOS ONE

Dear Dr. Salmon,

Thank you for submitting your manuscript to PLOS ONE. After careful consideration, we feel that it has merit but does not fully meet PLOS ONE’s publication criteria as it currently stands. Therefore, we invite you to submit a revised version of the manuscript that addresses the points raised during the review process.

We look forward to receiving your revised manuscript.

Kind regards,

Fausto Cavallaro, PhD

Academic Editor

PLOS ONE

Journal Requirements:

Additional Editor Comments: Dear Authors the reviewers ask for further improvement of your paper. Please address all their comments.

Reviewers' comments:

Reviewer's Responses to Questions

**Comments to the Author**

1. If the authors have adequately addressed your comments raised in a previous round of review and you feel that this manuscript is now acceptable for publication, you may indicate that here to bypass the “Comments to the Author” section, enter your conflict of interest statement in the “Confidential to Editor” section, and submit your "Accept" recommendation.

Reviewer #2: (No Response)

Reviewer #3: (No Response)

2. Is the manuscript technically sound, and do the data support the conclusions?

Reviewer #2: No

Reviewer #3: Partly

3. Has the statistical analysis been performed appropriately and rigorously? 

Reviewer #2: (No Response)

Reviewer #3: N/A

4. Have the authors made all data underlying the findings in their manuscript fully available?

Reviewer #2: Yes

Reviewer #3: Yes

5. Is the manuscript presented in an intelligible fashion and written in standard English?

Reviewer #2: Yes

Reviewer #3: Yes

6. Review Comments to the Author

Reviewer #2: The paper has improved. However, I still have significant issues with the manuscript, especially with the new parts.

1. Reading my comments you could have seen that I was talking about the ELECTRE Tri-B, and your response was that you used ELECTRE Tri-C, which is not accurate. If you have boundary profiles (Figure 2 for example is a place where you can see it), you are clearly talking about Tri-B. Please correct such mistakes.

2. The whole model setting is very problematic.

2.1. Please clearly define in the manuscript what are the alternatives, and more importantly criteria.

2.2. Biggest confusion causes Table 3. Are the values for the indifference, preference and veto threshold same for all wavelengths? One cannot understand it from the table.

3. I am not familiar with the concepts of indifference and non-compatibility in ELECTRE sorting models. Please clarify and check that part.

4. Please do a thorough proof reading before sending the manuscript. Page 5, line 100 you did not finish the name of the section; page 7, line 130, no dot at the end of the sentence; page 8, line 157 same thing. There are double spaces on several spots.

Finally, although improved, manuscript still has several sloppy mistakes and I urge authors to carefully read everything, including background methodological material and correct all the issues present in the current version of the manuscript.

Reviewer #3: The paper presents an interesting testing procedure to authenticate drugs applying a sorting algorithm based on ELECTRE TRI.

The problem is clearly stated, and the main contribution of the proposal is discussed. However, there are still some concerns with the sorting procedure proposed in the BSF test. Additionally, some statements in the manuscript are vague and could be rewritten. These statements are commented below.

# Line 82 – a multi-criteria sorting algorithm which can be used to compensate for the uncertainty when a pharmaceutical compound is analyzed “in the field” …

It is not clear that a multi-criteria sorting algorithm can be used to compensate for uncertainty. This terminology can be confusing in the decision analysis field.

As a suggestion, I would change the term uncertainty for imprecise , that is more appropriate.

# Line 91 - To compensate for samples prepared in the field using less than ideal equipment (i.e randomness in concentrations) the ELECTRE TRI-C multi-criteria sorting algorithm was adapted and applied to increase the likelihood of sorting a sample to the correct authentic/inauthentic category.

The sorting method presented is more related to the ELECTRE TRI-B, in which boundaries of the categories are defined. ELECTRE TRI-C does not depend on boundaries profiles but references examples. Please correct it in the manuscript.

I would not use the term “increase the likelihood of sorting a sample”, as the method ELECTRE was not designed for dealing with likelihood.

# Line 99-102 –difficult to read

# Line 149 – ELECTRE TRI-C is an algorithm that has specific attributes that allow for uncertainty and variability in collected data (or variability in sample preparation) that makes it ideal for the purposes of this method [17,18, 19, 20].

I disagree with this statement. Electre tri being the IDEAL method is very strong. The BSF is itself an interesting method to analyze the drugs. Besides the relevance of the studied problem, the sorting algorithm based on ELECTRE TRI does not seem to be the most appropriate one. (please consider that this comment does not invalidate the results). The point is, the study is using a sophisticated sorting technique that deals with the conflict of criteria and preferences of a decision-maker. However, the BSF problem seems to be more related to the calibration of measures by a specialist. The main point is that a similar result (dugs being considered authentic or not) could be reached by a procedure based on fuzzy logic. A Fuzzy method, including a similar calibration step, could be applied to define if a sample is authentic or not. Fuzzy systems (Zadeh, 1965) are well-known methods used to account for the knowledge of a specialist (in this case, the pharmaceutics). This is because a sample to be considered authentic must have all measures (for each wavelength of 240, 250, 260 and 270 – the criteria) under a specific value interval, but with some imprecise boundaries.

(observe that criteria weights are equal, and the credibility thresholds is 0,85)

Line 173 – mention ELECTRE TRI-C.

# Line 181 – Presents the sorting procedure.

In this case, the author could mention that an adapted procedure is used to sort the samples, based on the combined notion of the pessimistic and optimistic procedure of ELECTRE TRI-B. The explanation of left-to-right is interesting for a non-specialist in the area.

# Line - 282 Raw transmitted light absorbance data collected for 25 samples prepared in order to optimize ELECTRE TRI uncertainty parameters.

And

# Line 287. “This data displays three uncertainty parameters for ELECTRE TRI sorting algorithm. Values were selected by optimizing the minimize number of false positive and false negatives out of 50 samples of random substances (with 4 being factually positive)”.

It is not clear how the parameters have been optimized. This is quite important because the sorting procedure will depend on these established values.

In the response letter, the authors says that the solver function was applied to calibrate the categories. This should be mentioned in the text and explained in the supplement material, for instance.

# Line 350 - “There are a number of sorting algorithms that consider uncertainty in data, PROMETHEE II being one, that were also investigated.”

PROMETHEE II does not mean to sort alternatives.

# Line 361 - “Whereas ELECTRE TRI-C fails to sort this sample to any category, other methods would require that this be sorted to the best, if not correct, category. Thus, ELECTRE –TRI-C avoids the false positives challenge to a higher degree than other methods explored.”

I cannot see how ELECTRE can deal with false positives and negatives with such a sample. I don’t think this is the best argument. The procedure is not statistical.

There are some real applications of ELECTRE TRI method that could support this application. Consider Human Development Index Based on ELECTRE TRI Multicriteria Method, Social Indicators Research, 2016.

7. PLOS authors have the option to publish the peer review history of their article (what does this mean?). If published, this will include your full peer review and any attached files.

Reviewer #2: No

Reviewer #3: No

---

## [Author Response · Author response to Decision Letter 1]

21 Sep 2021

CMS response to REVIEWER #2

Reviewer #2: The paper has improved. However, I still have significant issues with the manuscript, especially with the new parts.

1. Reading my comments you could have seen that I was talking about the ELECTRE Tri-B, and your response was that you used ELECTRE Tri-C, which is not accurate. If you have boundary profiles (Figure 2 for example is a place where you can see it), you are clearly talking about Tri-B. Please correct such mistakes.

- Line 29 & Line 92 and throughout Updated to ELECTRE TRI B

2. The whole model setting is very problematic.

2.1. Please clearly define in the manuscript what are the alternatives, and more importantly criteria.

This is an unique and unfamiliar application of a sorting algorithm, and we agree with the reviewer that the reader might not be able to recognize the alternatives and criteria. Therefore, the following clarification will be added to the paper for clarity. We clarify that the alternatives:

Line 107 – 115 and continue to clarify thoughout the manuscript

- The purpose of multicriteria decision processes are to differentiate between alternatives through some analysis of the criteria used to describe or define each alternative. The mirage of available processes allows for this differentiation to be in the form of ranking, outranking, sorting , or clustering There are innumerable potential application of these methods that range from; a simple ranking of alternatives homes for purchase by criteria of cost, size, location, etc…, to compression of digital images by clustering pixels with similar characteristics.

The specific application in this research is for sorting samples of medications (the alternatives) into categories that differentiate (in this case, sort) the alternatives into authentic and inauthentic samples. The criteria used for this are four measures of absorbance captured at four wavelengths. Generally, alternatives are sorted using criteria of different metrics (weight, length, cost, etc…), but once these values of unique metrics are normalized to some non-dimensional scale, these are in effect assessed by a common unitless metric on some scale (0 to 1, 1 to 10, etc). In our application, the data collected from the device are normalized on a common unitless scale of absorbance 0 and 2, but at four different places on the Ultraviolet (UV) light spectrum. 

For this process, there are two sets of alternatives. The first set of 25 samples are used to calibrate the parameters of the ELECTRE TRI-B sorting algorithm. The second set of 50 are used to then test the calibrated sorting parameters. In both cases, the alternatives are the 25 and 50 prepared samples, and the criteria are the metrics of absorbance on four wavelengths in the UV spectrum.

2.2. Biggest confusion causes Table 3. Are the values for the indifference, preference and veto threshold same for all wavelengths? One cannot understand it from the table.

 Line 312 - 317 Clarified

- Yes, while it is possible that in future applications this will not be the case, for the specific application here, the preference thresholds were common across the four wavelengths to within any reasonable understanding of significant differences, including the indifference thresholds, which was near zero and so rounded to zero. Using rounded and common units across all four wavelengths had no impact on the results of sorting. It should be noted that future applications of this process might include 6 or more wavelengths either deeper into the UV spectrum or into visible light spectrum. Under these circumstances, it is possible, if not likely, that such a simplifying assumption will not hold, and one or all criteria (measures of absorbance at specific wavelengths) might need to be unique to that wavelength. This might also be the case for the four wavelengths of this application, but for different medications. However, preliminary study that has occurred since the initiate drafting of this manuscript suggests that common uncertainty parameters will suffice.

3. I am not familiar with the concepts of indifference and non-compatibility in ELECTRE sorting models. Please clarify and check that part.

A different paper might have used slightly different terms, but the source paper used for this application used these terms, and thus are repeated here. These terms are defined below directly from the source document based on the performance of the credibility score relative to the credibility threshold (M.E. Fontana, C.A, Cavalcante, 2011). The authors used different nomenclature than we use in the manuscript, but these concepts of indifference and incomparable are clear. As implemented: 1) indifference simply means that the algorithm does not definitively sort an alternative to a specific category, and thus is indifferent and leaves this to the decision maker, and 2) incomparable means that the sample simply cannot be sorted to any category. This is primarily because the alternative so fundamentally dissimilar to the logic of the category system that there is no meaningful placement possible. To be crude, it is like trying to sorting a table into a system designed for farm animials. Below is the authors’ of the sort document logic for preference, indifference and incomparability.

The values of σ(a, bh), σ(bh, a) ) and λ determine the preference situation between a and bh: 

The values of σ(a, bh), σ(bh, a) ) and λ determine the preference situation between a and bh: 

a) σ(a, bh)≥λ and σ(bh, a)≥λ → aSbh Band bhSa → aIbh, i. e., a is indifferent to bBhB; 

b) σ(a, bh)≥λ and σ(bh, a)<λ → aSbh and notbhSa → a ≻bh, i. e., a is preferred to bBh B(weakly or strongly); 

c) σ(a, bh)<λ and σ(bh, a)≥λ → not aSbh Band bhSa→ bh≻ a, i. e., bh Bis preferred to a (weakly or strongly); 

d) σ(a, bh)<λ and σ(bh, a)<λ → not aSbh Band not bhSa→ aRbh, i. e., a is incomparable to bh. 

4. Please do a thorough proof reading before sending the manuscript. Page 5, line 100 you did not finish the name of the section; page 7, line 130, no dot at the end of the sentence; page 8, line 157 same thing. There are double spaces on several spots.

- Line 73 - The Section has the correct name

- Line 130 - All sentence have periods at the end of it

- Line 157 - All sentence have periods at the end of it. You may be referring the Fig 1 legend which appears to blend with the main body of the manuscript due to the formatting required.

Finally, although improved, manuscript still has several sloppy mistakes and I urge authors to carefully read everything, including background methodological material and correct all the issues present in the current version of the manuscript.

We take exception to the reviewers reference to “sloppy mistakes.” This is a very complex method that we describe for a non mathematical global health audience in a very specific format. We are happy to change any errors you may point out. 

CMS response to REVIEWER #3

Reviewer #3: The paper presents an interesting testing procedure to authenticate drugs applying a sorting algorithm based on ELECTRE TRI.

The problem is clearly stated, and the main contribution of the proposal is discussed. However, there are still some concerns with the sorting procedure proposed in the BSF test. Additionally, some statements in the manuscript are vague and could be rewritten. These statements are commented below.

# Line 82 – a multi-criteria sorting algorithm which can be used to compensate for the uncertainty when a pharmaceutical compound is analyzed “in the field” …

It is not clear that a multi-criteria sorting algorithm can be used to compensate for uncertainty. This terminology can be confusing in the decision analysis field.

As a suggestion, I would change the term uncertainty for imprecise , that is more appropriate.

Line 357, 361, ……We agree, and the paper has been update. That said we did not find that “imprecise” was a clear term either and we therefore chose to use the word “randomness”

- 

 #Line 91 - To compensate for samples prepared in the field using less than ideal equipment (i.e randomness in concentrations) the ELECTRE TRI-C multi-criteria sorting algorithm was adapted and applied to increase the likelihood of sorting a sample to the correct authentic/inauthentic category.

The sorting method presented is more related to the ELECTRE TRI-B, in which boundaries of the categories are defined. ELECTRE TRI-C does not depend on boundaries profiles but references examples. Please correct it in the manuscript.

I would not use the term “increase the likelihood of sorting a sample”, as the method ELECTRE was not designed for dealing with likelihood.

- The manuscript has been updated as the reviewer suggests

# Line 99-102 –difficult to read

- We have changed the wording

# Line 149 – ELECTRE TRI-C is an algorithm that has specific attributes that allow for uncertainty and variability in collected data (or variability in sample preparation) that makes it ideal for the purposes of this method [17,18, 19, 20].

I disagree with this statement. Electre tri being the IDEAL method is very strong. The BSF is itself an interesting method to analyze the drugs. Besides the relevance of the studied problem, the sorting algorithm based on ELECTRE TRI does not seem to be the most appropriate one. (please consider that this comment does not invalidate the results). The point is, the study is using a sophisticated sorting technique that deals with the conflict of criteria and preferences of a decision-maker. However, the BSF problem seems to be more related to the calibration of measures by a specialist. The main point is that a similar result (dugs being considered authentic or not) could be reached by a procedure based on fuzzy logic. A Fuzzy method, including a similar calibration step, could be applied to define if a sample is authentic or not. Fuzzy systems (Zadeh, 1965) are well-known methods used to account for the knowledge of a specialist (in this case, the pharmaceutics). This is because a sample to be considered authentic must have all measures (for each wavelength of 240, 250, 260 and 270 – the criteria) under a specific value interval, but with some imprecise boundaries.

(observe that criteria weights are equal, and the credibility thresholds is 0,85)

- Line 172 - Rephrased as “convenient” rather than “ideal”. Further, we do not disagree with the reviewer, and future versions of this process will investigate using fuzzy logic as a possible approach. 

Line 173 – mention ELECTRE TRI-C.

- Added as per reviewers suggestion

- 

# Line 181 – Presents the sorting procedure.

In this case, the author could mention that an adapted procedure is used to sort the samples, based on the combined notion of the pessimistic and optimistic procedure of ELECTRE TRI-B. The explanation of left-to-right is interesting for a non-specialist in the area.

- Reviewed by authors for clarity

# Line - 282 Raw transmitted light absorbance data collected for 25 samples prepared in order to optimize ELECTRE TRI uncertainty parameters.

And

# Line 287. “This data displays three uncertainty parameters for ELECTRE TRI sorting algorithm. Values were selected by optimizing the minimize number of false positive and false negatives out of 50 samples of random substances (with 4 being factually positive)”.

It is not clear how the parameters have been optimized. This is quite important because the sorting procedure will depend on these established values.

In the response letter, the authors says that the solver function was applied to calibrate the categories. This should be mentioned in the text and explained in the supplement material, for instance.

- 

- This has been updated. There seems to have been some confusion between the authors as to what was main text and what was table legend. 

# Line 350 - “There are a number of sorting algorithms that consider uncertainty in data, PROMETHEE II being one, that were also investigated.”

PROMETHEE II does not mean to sort alternatives.

- This has been corrected as PROEMTHEE Sort (PROMSORT)

# Line 361 - “Whereas ELECTRE TRI-C fails to sort this sample to any category, other methods would require that this be sorted to the best, if not correct, category. Thus, ELECTRE –TRI-C avoids the false positives challenge to a higher degree than other methods explored.”

I cannot see how ELECTRE can deal with false positives and negatives with such a sample. I don’t think this is the best argument. The procedure is not statistical.

- This section has been reviewed and updated for clarity. However, to address the reviewers comment: the method was employed for this study specifically because it is not a statistical process. For any sorting system or sorting algorithm, there will be items that are sorted into a category based on the measured attributes. This sorting is not based on statistics, but on attributes of 1) the item/alternative, and 2) the defined parameters and boundaries of the system into which the item/alternative is being sorted. Further, there are instances wherein a sample cannot be sorted to any category simply because the attributes of the item/alternative is so fundamentally different than the system into which it is being sorted, that the system simply cannot accommodate the item. To put it crudely, it would be like sorting a chair into a system designed for sorting circus animals. ELECTRE TRI-B is convenient for use because for any sample that cannot be sorted (is incomparable to the system), we can simply assume that that specific sample/alternative/item is LQ/C (inauthentic) sample of the medication because it is so widely dissimilar to the sorting system and defined parameters. 

There are some real applications of ELECTRE TRI method that could support this application. Consider Human Development Index Based on ELECTRE TRI Multicriteria Method, Social Indicators Research, 2016.

---

## [Decision Letter · Decision Letter 2]

22 Nov 2021

PONE-D-20-14126R2Fake Drugs:  Using Baseline Spectral Fingerprinting and a sorting algorithm to infer quality of medicationsPLOS ONE

Dear Dr. Salmon,

Thank you for submitting your manuscript to PLOS ONE. After careful consideration, we feel that it has merit but does not fully meet PLOS ONE’s publication criteria as it currently stands. Therefore, we invite you to submit a revised version of the manuscript that addresses the points raised during the review process.

We look forward to receiving your revised manuscript.

Kind regards,

Fausto Cavallaro, PhD

Academic Editor

PLOS ONE

Journal Requirements:

Reviewers' comments:

Reviewer's Responses to Questions

**Comments to the Author**

1. If the authors have adequately addressed your comments raised in a previous round of review and you feel that this manuscript is now acceptable for publication, you may indicate that here to bypass the “Comments to the Author” section, enter your conflict of interest statement in the “Confidential to Editor” section, and submit your "Accept" recommendation.

Reviewer #2: (No Response)

Reviewer #3: All comments have been addressed

2. Is the manuscript technically sound, and do the data support the conclusions?

Reviewer #2: Partly

Reviewer #3: Yes

3. Has the statistical analysis been performed appropriately and rigorously? 

Reviewer #2: N/A

Reviewer #3: N/A

4. Have the authors made all data underlying the findings in their manuscript fully available?

Reviewer #2: Yes

Reviewer #3: Yes

5. Is the manuscript presented in an intelligible fashion and written in standard English?

Reviewer #2: Yes

Reviewer #3: Yes

6. Review Comments to the Author

Reviewer #2: The paper is improved. However, I still have a major concern regarding your main findings in Table 4. I raised the issue in the previous round of review, but the response from the authors is not adequate. There is no possibility that the sorted category for some samples is “Indiferent” or “Not Compatible”. You have three categories, CAT-A, CAT-B, and CAT-C. ELECTRE TRI methods sort alternatives using two approaches. You even use the reference in response (Fontana & Cavalcante, 2011), where everything is correctly stated. In that paper, the authors define these two approaches as pessimistic and optimistic.

What can happen is that ELECTRE provides an interval as the solution (even [CAT-A, CAT-C]). Still, your claim is not correct. We cannot have an “indifferent”, nor “not compatible” category. Also, if the sorted category is “indifferent” how come you claim that this represents correct sorting?

I have put my recommendation as a minor revision, but this is a major issue requiring correction.

I have spotted a missing dot, line 88, at the end of the sentence.

Reviewer #3: The authors have addressed most of the comments in a satisfactory way. Although the new version of the manuscript has been improved, some minor mistakes need to be corrected.

Considering the tracked version of the manuscript, it seems that a reference is missing:

Line 461 – “There are a number of sorting algorithms that consider randomness 461 uncertainty in data, PROMETHEE Sort being a relevant example (cite).” (??)

Another issue is the term “uncertainty parameters”. Please change “uncertainty parameter” to “threshold parameter”, which is more appropriate.

“uncertainty parameter” is used repeatedly along with the text. For instance, Line 305, line 311, Line 371, Line 443, and Line 456

Line 372 – The previous version was more appropriate: “To define the preference and veto threshold …”

In Electre methods, threshold are used to account for the imperfect knowledge of data, vagueness and some arbitrariness when building the criteria.

7. PLOS authors have the option to publish the peer review history of their article (what does this mean?). If published, this will include your full peer review and any attached files.

Reviewer #2: No

Reviewer #3: No

---

## [Author Response · Author response to Decision Letter 2]

24 Jan 2022

PLEASE REFERENCE OUR ATTACHED ' RESPONSE TO REVIEWERS' AS A BETTER EXPLANATION AND RESPONSE. WE HAVE INCLUDED MULTIPLE FIGURES WHICH WOULD NOT TRANSLATE INTO THE DIALOGUE BOX BELOW

6. Review Comments to the Author

Reviewer #2 Comments: 

The paper is improved. However, I still have a major concern regarding your main findings in Table 4. I raised the issue in the previous round of review, but the response from the authors is not adequate. There is no possibility that the sorted category for some samples is “Indiferent” or “Not Compatible”. You have three categories, CAT-A, CAT-B, and CAT-C. ELECTRE TRI methods sort alternatives using two approaches. You even use the reference in response (Fontana & Cavalcante, 2011), where everything is correctly stated. In that paper, the authors define these two approaches as pessimistic and optimistic.What can happen is that ELECTRE provides an interval as the solution (even [CAT-A, CAT-C]). Still, your claim is not correct. We cannot have an “indifferent”, nor “not compatible” category. Also, if the sorted category is “indifferent” how come you claim that this represents correct sorting? I have put my recommendation as a minor revision, but this is a major issue requiring correction. I have spotted a missing dot, line 88, at the end of the sentence.

Authors’ Response to Reviewer #2 comments

This is our response to Reviewer #2 comments about the sorting issue related to our application of ELECTRE TRI to sort samples of medications tested using ultraviolet light into categories of presumably authentic, low quality (LQ) and counterfeit (C) categories. The specific issue relates to Reviewer #2 challenging our reporting some samples using terms ‘indifferent’ and ‘incomparable’ rather than sorting each sample to a specific category of CAT-A (authentic), CAT-B (LQ/C) or CAT-C (LQ/C). We view this as a minor interpretive issue, which we describe below.

In short, we used a simple interpretation of results to sort samples that do not easily fit into the defined category system: 

• if any sample results in either an indifferent or incomparable result against any boundary, then that sample was defined as LQ/C. 

• for these samples, we reported the ‘indifferent’ or ‘incomparable’ result, rather than a category. 

• If we were to report a category, each would be sorted to either CAT-B or CAT-C, both of which are LQ/C

• We chose to report as Indifferent or Incomparable in order to discuss the potential to use this method to increase resolution of the analysis to further separate LQ versus C samples. 

Clarification of soring choices and reporting of results

To clarify our original reporting, we first capture an image of four logic statements (Statements A to D) from the paper by Fontana and Cavalcante, 2011, which are also reentered as text for clarity. Here, statements A and D clearly indicate that a sample can test as either indifferent and incomparable to a boundary. 

Figure 1: from Fantana and Cavalcante, 2011

Equations from paper recreated for clarity:

The values of σa,bℎ, σbℎ,aand λdetermine the preference situation between a and bℎ

a) σa,bℎ≥λ and σbℎ,a≥λ →aSbℎand bℎSa→aIbℎ, i.e., a is indifferent to bℎ 

b) σa,bℎ≥λ and σbℎ,a<λ→aSbℎand notbℎSa→a>bℎ, i.e., a is preferred to bℎ

c) σa,bℎ<λ and σbℎ,a≥λ→notaSbℎand bℎSa→bℎ>a i.e. bℎis preferred to a

d) σa,bℎ<λ and σbℎ,a<λ→notaSbℎand notbℎSa→bℎRa i.e. a is incomparable to bℎ

The Electra Tri process requires two ‘sweeps’ of a category system, starting on the left, with each sample being tested against the upper boundary of the left most category for each criterion. A credibility index (σa,bℎ) is calculated, and compared against a user defined threshold (λ). This is repeated for a right-to-left sweep of the sample against the lower boundary of the second category (which is the same as the upper boundary of the left most category). 

This is illustrated in Figure 2 as a graphical representation of the four possible outcomes determined by Statements A to D (Figure 1) against the first boundary of the category system. In this illustrations, the arrow on the left represents the results of the initial left-to-right sweep, and the arrow on the right represents the results of the right-to-left sweep. An illustration similar to this was used when describing the algorithm process with some authors, and is used here to further clarify the use of the method in our research. 

Figure 2: example of potential for four logic results for sample tested against first boundary 

Statement A:

- the left-to-right sweep suggests sample is superior to the boundary (arrow pushing right). However, the right-to-left sweep indicates the opposite (arrow pushing left). 

- This results in the situation wherein the algorithm is ‘indifferent’ as to which category an item is sorted. We, therefore, defined the item as LQ/C and reported simply as “indifferent”.

- Specific examples reported in our paper include samples 1, 33 and 46. Each of these samples were purposefully designed a ‘proxy low quality’ samples to test the resolution of the method. 

- The indifferent result reported demonstrates the potential for using our process for differentiating between LQ and C samples. We did not fully exploit this in our paper because we did not feel that the preliminary data support this level of resolution, but we intend to return to the research with more data in the future. 

 Statement B: 

- Both the left and right sweeps indicate that Sample is superior to the boundary (both arrows pushing right), and thus the sample is preliminarily sorted to CAT-A.

- This Sample must be subsequently tested in an identical fashion against the next boundary between Cat-A and Cat-C. 

- When tested against this next boundary, all four outcomes are again possible, and a sample might be indifferent (LQ/C), superior (CAT-C), inferior (CAT-A) or incomparable (LQ/C) to the boundary.

Statement C:

- both the left and right sweep are in agreement that the sample is sorted to CAT-B (arrows pushing left), and no further analysis is required

Statement D:

- This situation takes some interpretation because the results are not immediately clear. Figure 3 clarifies the issue with an illustrative example of an extreme example wherein such a situation can result. 

- Here, there are no absorbance data factually located in the central category (CAT-A), and thus the sample cannot be sorted to CAT-A. Further, given that only half the data are located in CAT-B, the sample cannot be sorted to CAT-B, with a similar situation for CAT-C. 

- The assessment of this sample against either boundary results in an incomparable result (Statement D)

- For our research, any item defined as incomparable to any boundary was simply defined as LQ/C and reported as incomparable. 

Closing

We agree that reporting results as indifferent or incomparable is not an absolutely strict application of the ELECTRE TRI method. However, we also feel that this slight deviation of application by reporting results as indifferent or incomparable is well within the realm of reasonable extrapolation and application of the algorithm. 

The purpose of reporting results the way we did was simply to prepare for future research wherein we may be able to leverage the power of the algorithm to differentiate between LQ and C, as we did in the current research by noting that the algorithm was able to detect purposefully prepared LQ samples (which we reported as indifferent). 

We intend to continue this research with larger sample sets in the future using field procured samples of medications in low and middle income countries, at which point we will also be experimenting with a 5 category system, with a low quality category located between the current CAT-B to CAT-A boundary and CAT-A to CAT-C boundary that take advantage of the ability to differentiate between LQ and C samples

---

## [Decision Letter · Decision Letter 3]

21 Apr 2022

PONE-D-20-14126R3Fake Drugs:  Using Baseline Spectral Fingerprinting and a sorting algorithm to infer quality of medicationsPLOS ONE

Dear Dr. Salmon,

Thank you for submitting your manuscript to PLOS ONE. After careful consideration, we feel that it has merit but does not fully meet PLOS ONE’s publication criteria as it currently stands. Therefore, we invite you to submit a revised version of the manuscript that addresses the points raised during the review process.

We look forward to receiving your revised manuscript.

Kind regards,

Fausto Cavallaro, PhD

Academic Editor

PLOS ONE

Additional Editor Comments: I strongly suggest to reply point by point to Reviewers 2's comments

Reviewers' comments:

Reviewer's Responses to Questions

**Comments to the Author**

1. If the authors have adequately addressed your comments raised in a previous round of review and you feel that this manuscript is now acceptable for publication, you may indicate that here to bypass the “Comments to the Author” section, enter your conflict of interest statement in the “Confidential to Editor” section, and submit your "Accept" recommendation.

Reviewer #2: (No Response)

Reviewer #3: All comments have been addressed

Reviewer #4: (No Response)

2. Is the manuscript technically sound, and do the data support the conclusions?

Reviewer #2: No

Reviewer #3: Yes

Reviewer #4: Partly

3. Has the statistical analysis been performed appropriately and rigorously? 

Reviewer #2: N/A

Reviewer #3: N/A

Reviewer #4: No

4. Have the authors made all data underlying the findings in their manuscript fully available?

Reviewer #2: Yes

Reviewer #3: Yes

Reviewer #4: Yes

5. Is the manuscript presented in an intelligible fashion and written in standard English?

Reviewer #2: (No Response)

Reviewer #3: Yes

Reviewer #4: Yes

6. Review Comments to the Author

Reviewer #2: I thank the authors for an attempt to clarify the reasoning. Several critical issues emerged from the response.

1. What authors refer to in Fantana and Cavalcante's article are ways to obtain pairwise relations between two alternatives. This has nothing to do with the pessimistic and optimistic rules where you actually have the major problem. Each rule defines a category with certainty, i.e., the upper and lower boundary of the sorting procedure.

2. Note that you do not have this reference in the reference list in the manuscript. Generally, you have mainly missed the good references in presenting ELECTRE Tri-B method, even though you had good suggestions from reviewers.

Overall, the idea behind the work is good. The idea behind the method is very solid, and it can work well. The execution is poor. I was very sympathetic toward the authors, addressing the main issues several times, but this has still not improved. In the end, you remain with the clear methodological mistake of using indifferent and not compatible samples and evaluating them as correct classification.

Reviewer #3: The authors have justified the proposed sorting procedure applied in the paper, based on the notions of ELCTRE-TRI-B.

In this way, the authors should avoid generalizations regargind applicarons of ELECTRE-TRI. For instance, I would recommend changing the following sentence (line 236, pg 10): "In the general application of ELECTRE TRI-B this situation defines Sample-Y as being defined as “incomparable” to the sorting system. In other words, the sample cannot logically be sorting to any category of the sorting system as defined by the decision maker" to: "This situation defines Sample-Y as being defined as “incomparable” to the sorting system"

Reviewer #4: This paper presents a novel method for inferring quality of medications specifically.The current study represents an interesting subject to improve the healthcare system. However, the authors have to detail the results. Some comments have to be revised.

1- Authors have integrated the multi-criteria tool for sorting. However, the theoretical content is missing in the paper. It would be preferable to include a section about the multi-criteria tools.

2- The authors should present a comparative study of the multi-criteria sorting tools.

3- There are several multi-criteria sorting algorithms that can realize the study. The authors have chosen the use of ELECTRE TRI-B. They should justify the choice of this sorting algorithm.

4- To prove this classification proposed by the authors, it is preferable to study the sensibility analysis according to the cut-value λ.

7. PLOS authors have the option to publish the peer review history of their article (what does this mean?). If published, this will include your full peer review and any attached files.

Reviewer #2: No

Reviewer #3: No

Reviewer #4: **Yes: **Layla AZIZ

---

## [Author Response · Author response to Decision Letter 3]

9 Oct 2022

Authors response is presented in bold 

Reviewer #2

● I thank the authors for an attempt to clarify the reasoning. Several critical issues emerged from the response. 

o 1. What authors refer to in Fantana and Cavalcante's article are ways to obtain pairwise relations between two alternatives. This has nothing to do with the pessimistic and optimistic rules where you actually have the major problem. Each rule defines a category with certainty, i.e., the upper and lower boundary of the sorting procedure. 

▪ There are two alternatives. There is the sample (alternative 1) and the boundary between categories (proxy alternative). And the whole thing is a pair comparison between alternatives (samples) and category boundaries. We no longer use the nomenclature of “pessimistic and optimistic.” 

o 2. Note that you do not have this reference in the reference list in the manuscript. Generally, you have mainly missed the good references in presenting ELECTRE Tri-B method, even though you had good suggestions from reviewers. 

▪ The Fantana Cavalcante’s paper was used in a previous response to the reviewer’s comments, as it was not foundational to the research or manuscript. If the reviewer is requesting a particular citation, please so state. Most likely, we would be happy to accommodate.

o Overall, the idea behind the work is good. The idea behind the method is very solid, and it can work well. The execution is poor. I was very sympathetic toward the authors, addressing the main issues several times, but this has still not improved. In the end, you remain with the clear methodological mistake of using indifferent and not compatible samples and evaluating them as correct classification.

▪ 

The authors fundamentally disagree with the contention that indifferent and not compatible samples are not evaluated correctly as LQ/C. By definition samples that are not sorted to a specific category are LQ/C. We stated this previously, and have updated the paper to emphasis this point. This is literally why we selected the ELECTRE TRI process.

Reviewer #3

● The authors have justified the proposed sorting procedure applied in the paper, based on the notions of ELCTRE-TRI-B. In this way, the authors should avoid generalizations regarding applications of ELECTRE-TRI. For instance, I would recommend changing the following sentence (line 236, pg 10): "In the general application of ELECTRE TRI-B this situation defines Sample-Y as being defined as “incomparable” to the sorting system. In other words, the sample cannot logically be sorting to any category of the sorting system as defined by the decision maker" to: "This situation defines Sample-Y as being defined as “incomparable” to the sorting system" 

o 229-248 We appreciate the reviewer’s constructive guidance, and we believe that we have updates the language to satisfy their concern

Reviewer #4: 

● This paper presents a novel method for inferring quality of medications specifically. The current study represents an interesting subject to improve the healthcare system. However, the authors have to detail the results. Some comments have to be revised. 

o 1- Authors have integrated the multi-criteria tool for sorting. However, the theoretical content is missing in the paper. It would be preferable to include a section about the multi-criteria tools. 

▪ We appreciate the observation, and have included the underlying algorithm as a supplemental 

o 2- The authors should present a comparative study of the multi-criteria sorting tools. 

▪ 

We disagree that it is necessary to include analysis using competitor algorithms, if for no other reason than ‘which ones”? there are scores of approaches that could be used, including other MCDA oriented methods, as well as pattern recognition software (including Artificial Neural Networks). We selected the ELECTRE TRI-B method through consultation with normative experts, and used this algorithm based on their recommendations. Repeating the analysis using other methods might be informative from an abstract conceptual learning context to the MCDA community. But the purposes here is to document the results of an effective process, and as such determining if other methods are as effective or not is a distraction

o 3- There are several multi-criteria sorting algorithms that can realize the study. The authors have chosen the use of ELECTRE TRI-B. They should justify the choice of this sorting algorithm. 

▪ 

See note above as to why the ELECTRE TRI-B algorithm was selected. In addition, we updated the paper to make this point clearer to the reader. But in short: it was recommended, we tried it and it worked. Future work can return to this issue as a comparative analysis of various methods (including non-MCDA methods) as an interesting follow up body of research, but this work currently stands as it is.

o 4- To prove this classification proposed by the authors, it is preferable to study the sensibility analysis according to the cut-value λ. 

▪ 

We agree that this could be an interesting addition to the paper, and λ was adjusted to see impact on results. Lowering to 0.75 had no material impact on the results because anything that is not sorted to CAT-A is by definition LQ/C, and thus everything was still correctly sorted. However, by increasing to 0.87 one Authentic sample was reclassified as LQ/C. This is an interesting observation but still speaks to the benefit of using the preliminary 25 samples to train the model parameters, and that this was an effective way to define the parameter values.

---

## [Decision Letter · Decision Letter 4]

3 Nov 2022

PONE-D-20-14126R4Fake Drugs:  Using Baseline Spectral Fingerprinting and a sorting algorithm to infer quality of medicationsPLOS ONE

Dear Dr. Salmon,

Thank you for submitting your manuscript to PLOS ONE. After careful consideration, we feel that it has merit but does not fully meet PLOS ONE’s publication criteria as it currently stands. Therefore, we invite you to submit a revised version of the manuscript that addresses the points raised during the review process.

We look forward to receiving your revised manuscript.

Kind regards,

Fausto Cavallaro, PhD

Academic Editor

PLOS ONE

Additional Editor Comments:

The reviewer asserted that his comments are not addressed. Please proceed to revise carefully your paper on the base of his suggestions.

Reviewers' comments:

Reviewer's Responses to Questions

**Comments to the Author**

1. If the authors have adequately addressed your comments raised in a previous round of review and you feel that this manuscript is now acceptable for publication, you may indicate that here to bypass the “Comments to the Author” section, enter your conflict of interest statement in the “Confidential to Editor” section, and submit your "Accept" recommendation.

Reviewer #4: (No Response)

2. Is the manuscript technically sound, and do the data support the conclusions?

Reviewer #4: Partly

3. Has the statistical analysis been performed appropriately and rigorously? 

Reviewer #4: No

4. Have the authors made all data underlying the findings in their manuscript fully available?

Reviewer #4: Yes

5. Is the manuscript presented in an intelligible fashion and written in standard English?

Reviewer #4: Yes

6. Review Comments to the Author

Reviewer #4: Tha authors have not revised the fourth comment(To prove this classification proposed by the authors, it is preferable to study the sensibility analysis according to the cut-value λ.). The authors are asked to add a new section to include the sensitivity analysis study.

7. PLOS authors have the option to publish the peer review history of their article (what does this mean?). If published, this will include your full peer review and any attached files.

Reviewer #4: No

---

## [Author Response · Author response to Decision Letter 4]

23 Dec 2022

Review #4 Comment

The authors have not revised the fourth comment (To prove this classification proposed by the authors, it is preferable to study the sensibility analysis according to the cut-value λ.). The authors are asked to add a new section to include the sensitivity analysis study.

Author Response:

We had included a discussion on sensitivity of the cut-value / credibility threshold in the last version, but it was located in the discussion section. This has been updated with more information and moved to the end of the proof of concept section. 

Line 641-651 & 765-774

---

## [Decision Letter · Decision Letter 5]

24 Jan 2023

Fake Drugs:  Using Baseline Spectral Fingerprinting and a sorting algorithm to infer quality of medications

PONE-D-20-14126R5

Dear Dr. Salmon,

We’re pleased to inform you that your manuscript has been judged scientifically suitable for publication and will be formally accepted for publication once it meets all outstanding technical requirements.

Kind regards,

Fausto Cavallaro, PhD

Academic Editor

PLOS ONE

**Comments to the Author**

1. If the authors have adequately addressed your comments raised in a previous round of review and you feel that this manuscript is now acceptable for publication, you may indicate that here to bypass the “Comments to the Author” section, enter your conflict of interest statement in the “Confidential to Editor” section, and submit your "Accept" recommendation.

Reviewer #4: All comments have been addressed

2. Is the manuscript technically sound, and do the data support the conclusions?

Reviewer #4: Yes

3. Has the statistical analysis been performed appropriately and rigorously? 

Reviewer #4: Yes

4. Have the authors made all data underlying the findings in their manuscript fully available?

Reviewer #4: Yes

5. Is the manuscript presented in an intelligible fashion and written in standard English?

Reviewer #4: Yes

6. Review Comments to the Author

Reviewer #4: (No Response)

7. PLOS authors have the option to publish the peer review history of their article (what does this mean?). If published, this will include your full peer review and any attached files.

Reviewer #4: **Yes: **layla aziz

---

## [Editor Report · Acceptance letter]

3 Apr 2023

PONE-D-20-14126R5 

Fake Drugs:  Using Baseline Spectral Fingerprinting and a sorting algorithm to infer quality of medications 

Dear Dr. Salmon:

I'm pleased to inform you that your manuscript has been deemed suitable for publication in PLOS ONE. Congratulations! Your manuscript is now with our production department. 

Kind regards, 

on behalf of

Professor Fausto Cavallaro 

Academic Editor

PLOS ONE